# Genetic basis for coordination of meiosis and sexual structure maturation in *Cryptococcus neoformans*

Linxia Liu[1,2†], Guang-Jun He[1†], Lei Chen[1,2†], Jiao Zheng[3], Yingying Chen[1,2], Lan Shen[1], Xiuyun Tian[1], Erwei Li[1], Ence Yang[4], Guojian Liao[3], Linqi Wang[1,2*]

[1]State Key Laboratory of Mycology, Institute of Microbiology, Chinese Academy of Sciences, Beijing, China; [2]University of Chinese Academy of Sciences, Beijing, China; [3]College of Pharmaceutical Sciences, Southwest University, Chongqing, China; [4]Department of Microbiology, School of Basic Medical Sciences, Peking University Health Science Center, Beijing, China

**Abstract** In the human fungal pathogen *Cryptococcus neoformans*, sex can benefit its pathogenicity through production of meiospores, which are believed to offer both physical and meiosis-created lineage advantages for its infections. *Cryptococcus* sporulation occurs following two parallel events, meiosis and differentiation of the basidium, the characteristic sexual structure of the basidiomycetes. However, the circuit integrating these events to ensure subsequent sporulation is unclear. Here, we show the spatiotemporal coordination of meiosis and basidial maturation by visualizing event-specific molecules in developing basidia defined by a quantitative approach. Monitoring of gene induction timing together with genetic analysis reveals co-regulation of the coordinated events by a shared regulatory program. Two RRM family regulators, Csa1 and Csa2, are crucial components that bridge meiosis and basidial maturation, further determining sporulation. We propose that the regulatory coordination of meiosis and basidial development serves as a determinant underlying the production of infectious meiospores in *C. neoformans*.
DOI: https://doi.org/10.7554/eLife.38683.001

**\*For correspondence:**
wanglq@im.ac.cn

[†]These authors contributed equally to this work

**Competing interests:** The authors declare that no competing interests exist.

## Introduction

Sex is pervasive throughout eukaryotes, including fungi. In the ubiquitous human fungal pathogen *Cryptococcus neoformans*, a model organism for fungal sex studies, sexual reproduction is considered to play an important role in promoting its infections (*Idnurm et al., 2005*; *Heitman et al., 2014*). For instance, sexual spores in *C. neoformans* are presumed infectious particles due to their special physical features, including oxidative stress resistance and small size, which enables compatible deposition in the pulmonary alveoli after inhalation (*Giles et al., 2009*; *Velagapudi et al., 2009*; *Botts and Hull, 2010*; *Kozubowski and Heitman, 2012*; *Ballou and Johnston, 2017*). Notably, sporulation in *C. neoformans* has not been observed during the mitotic life cycle under laboratory condition or in nature but is exclusively associated with sexual (meiotic) reproduction (*Kozubowski and Heitman, 2012*; *Huang and Hull, 2017*). This feature is mechanistically different from that of many human fungal pathogens in which asexual reproduction serves as the major route to produce genetically identical spore progenies (*Huang and Hull, 2017*). By comparison, due to meiotic recombination, meiospore progenies appear to have more diverse genomes, thereby potentially providing a lineage benefit associated with *Cryptococcus* infections and drug resistance (*Ni et al., 2013*).

*C. neoformans* has two defined sexual programs underlying sporulation, bisexual and unisexual reproduction (also named haploid fruiting) (*Kwon-Chung, 1976*; *Lin et al., 2005*; *Wang and Lin,*

**eLife digest**  Many microbes that cause disease form spores to survive during and between infections. These include the fungus *Cryptococcus neoformans*, which is the leading cause of fungal meningitis worldwide. This fungus produces spores via sexual reproduction, meaning the genes from two living strains of the fungi combine to create new lives with unique genetics. By diversifying the fungus's genetics, sexual reproduction in *Cryptococcus* is considered to accelerate drug resistance.

Several processes must be coordinated for *Cryptococcus* to reproduce sexually. Genetic information recombines through a process called meiosis, the spore-making cell (known as the sexual structure) matures and later spores are produced. Scientists have identified many genes involved in each of these processes. Yet it is not known how these processes are coordinated to ensure the proper sequence of events.

Liu, He, Chen et al. studied the physical changes in *Cryptococcus* cells when they lost certain genes. Two genes, which the researchers named *CSA1* and *CSA2*, were found to regulate the parallel progression of meiosis and maturation of the sexual structure. Both processes need to be complete before spore production begins. Further investigation showed that these genes are important across various strains of infectious *Cryptococcus*.

This research highlights sexual reproduction as a target to stop *Cryptococcus* forming spores and starting infections. The results also show that these processes change little through evolution within a large group of fungi. The next step will be to see how these systems operate across species and the effect this has on spore production.

DOI: https://doi.org/10.7554/eLife.38683.002

*2011*; *Fu et al., 2015*). Bisexual reproduction occurs between cells from two opposite mating types (α and a) (*Kwon-Chung, 1976*), while unisex only involves cells from a single mating type (mostly α) (*Lin et al., 2005*). Both reproduction modes involve similar sequential morphological differentiation and molecular events (*Fu et al., 2015*) (*Figure 1A*). These sexual cycles are initiated by the mating MAPK pathway in response to mating-inducing signals. The activated mating cascade subsequently induces a transition of a subpopulation of yeast cells in the mating community to hyphae, including invasive and aerial hyphae (*Wang et al., 2014*). Aerial hyphae stochastically differentiate into basidia (also named fruiting bodies) on their apexes, which represent a hallmark sexual structure of the phylum Basidiomycota. In many basidiomycetes including *C. neoformans*, meiosis usually occurs and progresses within basidia, leading to production of four meiotic products (*Kües, 2000*; *Wang et al., 2014*). The meiotic products undergo repeated rounds of mitosis and yield multiple nuclei that are packaged into meiospores (basidiospores), ultimately generating four chains of spores from the tip of the basidium (*Fu et al., 2015*). As two pre-sporulation events, the formation of basidia and the meiotic cycle individually offer a developmental basis and genetically distinct genomes for the production of meiospores. Thus, cryptococcal sporulation likely requires successful integration of these two events during sexual development (*Figure 1A*). This hypothesis remains to be proven due to the fact that the genetic basis for orchestrating meiosis and basidial differentiation remains poorly understood in *C. neoformans* and other basidiomycetes.

In this work, we developed a quantitative phenotypic method and identified a novel basidium indicator molecule to define the various stages during basidial differentiation. Using these approaches, we confirmed that basidial maturation and the meiotic cycle are spatiotemporally coordinated in *C. neoformans*. By profiling gene induction during mating differentiation, we further revealed that the coordination of these events is likely attributed to integrated control mediated by a shared regulatory program. Two RNA recognition motif (RRM) family RNA-binding proteins, Csa1 and Csa2, are the central members that specifically orchestrate meiosis and basidial differentiation, further directing sporulation. Together, our findings provide important insight into how the regulatory coordination of meiosis and basidial development is genetically determined to ensure the formation of resistant meiospores, which are predicted to have both physical and lineage advantages benefiting *Cryptococcus* infections.



**Figure 1.** Basidial differentiation and meiotic progression are spatiotemporally coordinated in *Cryptococcus neoformans*. (**A**) Diagram depicting hyphal development and meiosis-concomitant differentiation process in *C. neoformans*. (**B**) Schematic diagram outlining the basidial maturation score (BMS). (**C**) Violin plot analysis indicates that populations of basidia with high BMS gradually increased over time during unisexual and bisexual development. XL280α cells alone (unisex) or a mixture of XL280α and XL280**a** cells (α-**a** bisex) were dropped onto V8 medium and incubated at 25°C in the dark. Hyphae with or without basidia were photographed at 2, 4, 7 and 14 days after mating stimulation, and were randomly chosen for the BMS calculation (*n* > 110 for each time point). (**D**) Basidia were photographed at 7 days after mating stimulation. Unisex: *n* = 194 (unsporulated basidia); *n* = 51 (sporulated basidia). Bisex: *n* = 163 (unsporulated basidia); *n* = 53 (sporulated basidia). Bin width = 0.2. \*\*\*p<0.001, Kolmogorov-Smirnov test, two sided. A BMS of 1.0 was arbitrarily set as the threshold to define the basidial state. (**E**) Dynamic fluorescent intensity of Dmc1-mCherry during basidial maturation defined by the BMS method in cryptococcal unisex and bisex processes, respectively. *n* > 20 for each BMS range.

DOI: https://doi.org/10.7554/eLife.38683.003

The following source data and figure supplements are available for figure 1:

**Source data 1.** Source file for *Figure 1C,D and E*.
DOI: https://doi.org/10.7554/eLife.38683.006

**Figure supplement 1.** Violin plots showing the distribution frequency of the BMS in different strains.
DOI: https://doi.org/10.7554/eLife.38683.004

**Figure supplement 1—source data 1.** Source file for *Figure 1—figure supplement 1*.
DOI: https://doi.org/10.7554/eLife.38683.005

## Results

### Basidial maturation and meiotic progression are spatiotemporally coordinated in *C. neoformans*

The cellular features of the different phases during basidial maturation in basidiomycetes are poorly understood due to the lack of methods to define them. In this regard, we developed a criterion (BMS: basidial maturation score) to quantitatively evaluate basidial maturation using the ratio of the diameters between the basidium and its connected hyphae (*Figure 1B*). In both reproduction modes, the average BMS level gradually increased over time (*Figure 1C*). This result reflects a dynamic enlargement of basidia during sexual reproduction. In *C. neoformans*, spores are produced at the tip of basidia after the completion of fruiting body differentiation. Thus, sporulated basidia represent the mature state of basidial development. The BMS of sporulated basidia was significantly higher than that of un-sporulated basidia (p<0.0001, Kolmogorov-Smirnov test, two sided) (*Figure 1D*). Moreover, there was no evident change in the average BMS in the sporulated basidia population throughout both unisexual and bisexual development (data not shown). This finding suggests that, as the final state of basidial maturation, sporulated basidia cannot undergo further enlargement. These data indicate that BMS analysis represents a reliable approach for evaluating basidial maturation in *C. neoformans*. In this study, a BMS of 1.6 was set as the threshold to define the mature state or late stage of fruiting body development, as all the sporulated basidia tested showed a BMS over 1.6 (*Figure 1D*).

Next, we sought to evaluate the meiosis activity at different stages during basidial maturation, which were assessed by the BMS method. The meiosis-specific recombinase, Dmc1, with basidium-specific expression, has been employed as a molecular indicator of meiosis, due to its conserved function in meiotic recombination among divergent eukaryotes and high abundance during the meiotic cycle (*Lin et al., 2005*; *Wang et al., 2014*). The fluorescent signal of mCherry-fused Dmc1 was measured throughout basidial maturation. Interestingly, in both sexual cycles, Dmc1 displayed similar expression patterns during basidial maturation ($r = 0.84$, p<0.05, Pearson's test). Using a quantitative fluorescence assay, Dmc1 expression was first detected after basidial initiation and dramatically increased during the enlargement of basidia from BMS 1.2 to 1.6, while it began to decline as the BMS exceeded 1.6 (*Figure 1E*). These data indicate an explicit correlation between basidial maturation and meiosis-specific gene expression, supporting the hypothesis that meiotic progression and basidial differentiation are spatiotemporally coordinated in *C. neoformans*.

### Meiotic progression and basidial maturation are genetically associated

Next, we questioned whether meiotic progression and basidial maturation are genetically associated in *C. neoformans* and whether this association ensures their coordination in space and time. To tackle this question, we first investigated the effect of meiosis-specific components (Dmc1 and Spo11) on basidial maturation (*Lin et al., 2005*; *Feretzaki and Heitman, 2013*). Compared with the wild-type strain, the *dmc1Δ* and *spo11Δ* mutant showed only a slightly increased population of large basidia during unisexual reproduction (*Figure 1—figure supplement 1*). This finding indicates that the absence of these meiosis essential genes cannot impair basidial maturation. Thus, basidial differentiation can be achieved independently of meiotic progression per se. Instead, the coordination of the meiotic cycle and basidial differentiation may potentially be attributed to integrated control by a shared regulatory program that orchestrates these events. Our previous study has shown that the Pumilio family RNA binding protein Pum1 is important for the expression of the meiosis protein Dmc1 and post-meiotic sporulation (*Wang et al., 2014*; *Kaur and Panepinto, 2016*). This suggested us to test whether Pum1 is also involved in basidial maturation. A violin-plot analysis revealed that disruption of *PUM1* led to a significant decrease in mature basidial populations during both unisexual and bisexual development (95% CI: 0.02, 0.16, p=0.01 for unisex; 95% CI: 0.10, 0.29, p=9.1 × $10^{-5}$ for bisex; two tailed Student's-*t* test) (*Figure 2A*). Considering Pum1's contribution to basidial maturation, we speculated that Pum1 may play an important regulatory role in modulating the expression of the proteins residing in basidia. During sexual development, a highly dynamic expression has been observed in many genes encoding secretory or cell surface proteins, which contain signal peptide destined towards the secretory pathway. Some of these proteins exhibit specificity for enrichment in disparate morphotypes (yeast, hypha and basidium) from the *Cryptococcus*



**Figure 2.** Pum1 orchestrates meiotic progression and basidial maturation. (A) Violin plot analysis shows that disruption of *PUM1* cascade members led to a decrease of high BMS basidial population during both unisexual and bisexual development (*n* = 150 for each strain). (B) Morphotype-specific enrichment of Dha1, Fas1, and Fad1. >50 cells in each morphotype were examined for mCherry-labelled proteins expression. (C) Dynamics of Fad1-mCherry expression during unisexual development. Fad1-mCherry shows a remarkably biased expression in the basidium structure and displays different localization patterns. Scale bar: 5 μm. (D) Cells were placed onto a V8 plate at 25°C in the dark for unisexual induction, and incubated for 7 days. For each BMS range, >20 basidia expressing Fad1-mCherry were examined. The right panel highlights the dynamic enrichment of patterns I, II and IV at various stages during basidial maturation. (E) Predominant Fad1 protein exhibited a subcellular localization identical to pattern IV in post-meiotic basidia (sporulated basidia) during unisexual reproduction. Thirty-seven sporulated basidia expressing Fad1-mCherry were measured. ND = Not Detected. (F) Sporulation phenotypes for wild-type XL280α, the *fad1Δ* deletion mutant (unisexual reproduction), a wild-type cross between XL280α and XL280**a**, and the *fad1Δ* bilateral mutant cross. Scale bar: 10 μm (upper and middle panels), 5 μm (bottom panels).

DOI: https://doi.org/10.7554/eLife.38683.007

The following source data and figure supplement are available for figure 2:

**Source data 1.** Source file for *Figure 2A,D and E*.
DOI: https://doi.org/10.7554/eLife.38683.009

**Figure supplement 1.** Morphotype-specific expression patterns of Dha1, Fas1 and Fad1, which are fused by mCherry at their C-terminus.
DOI: https://doi.org/10.7554/eLife.38683.008

mating community (*Wang et al., 2014*). In a previous microarray study, Pum1 strongly upregulated the expression of multiple genes, whose products contain signal peptides predicted using SignalP and WoLF PSORT programs. These products include Fas1, Dha1 and Fad1 (*Figure 2B*) (*Wang et al., 2014*). The ample expression of mCherry-labeled Dha1 and Fas1 has been visualized in the fruiting body, while they can also be clearly detected in other morphotypes (*Wang et al., 2014*; *Gyawali et al., 2017*). These findings were recapitulated by our fluorescence microscopy analysis (*Figure 2B* and *Figure 2—figure supplement 1*). This result led us to examine whether Fad1, another important target of Pum1, displays a similar basidium-enriched expression feature. Indeed, we observed abundant Fad1 in almost all basidia examined. However, in contrast to Dha1 and Fas1, which can be strongly expressed in other morphotypes in addition to basidium, Fad1-mCherry could not be observed in yeast cells and most hyphae (*Figure 2B* and *Figure 2—figure supplement 1*). Only a small hyphal population appeared to very weakly express this protein (*Figure 2—figure supplement 1*). This finding indicates that Fad1 functions as a basidium-enriched protein. In basidia, this protein displayed four major subcellular localization patterns (*Figure 2B and C*). To our surprise, these patterns were highly dynamic during basidial maturation and were strongly correlated with specific basidial stages. For instance, as basidial development began, Fad1 was observed nearly exclusively in a diffuse form throughout the basidial cytoplasm (pattern I) or a condensed form as foci-like structures (pattern II). Subsequently, Fad1 accumulated on the 'neck' region connecting the basidia and hyphae (pattern III) during basidial enlargement, and was eventually localized to the surface of the upper region of the fruiting body (pattern IV), which was predominant in sporulated basidia or large fruiting body populations with a BMS above 1.6 (*Figure 2D and E*). Considering that the localization patterns of Fad1 are greatly related to specific basidial phases, Fad1 could be used as an indicator to define the various phases during basidial maturation in *C. neoformans*.

Given that Fad1 is enriched in the basidium, Fad1 may play a role in fruiting body differentiation or subsequent sporulation. The detailed phenotypic assays indicated that disrupting *FAD1* cannot compromise basidial maturation or morphogenesis but indeed perturbed post-meiotic sporulation. In both reproduction modes, shorter spore chains were produced in the *fad1Δ* mutant, in which spores from different chains adhered to each other, leading to intertwined spore chains coiling at the top of the basidia (*Figure 2F*). Thus, Fad1 is required for proper sporulation and spore dispersal.

## Monitoring gene induction timing during unisexual development reveals the gene network orchestrating meiosis and basidium development

In addition to meiosis and basidial differentiation, Pum1 is also involved in other mating processes, such as filamentation and α-a cell-cell fusion (*Wang et al., 2014*). Thus, Pum1 appears to act as a pleiotropic regulator that coordinates the sequential stages of sexual development rather than specifically connecting meiosis and fruiting body differentiation. To reveal the regulatory program that specifically and critically integrates the meiotic cycle and basidial maturation, we employed a high-coverage strand-specific RNA-sequencing analysis to compare whole-genome expression at five time points (6 hr, 12 hr, 24 hr, 48 hr and 72 hr) throughout unisexual development in the XL280 background (*Figure 3A*). No later time point was tested because a certain number of basidia have been formed at 72 hr post-mating stimulation (*Figure 1C*), and the genes activating this process should be transcriptionally induced earlier. We obtained 8177 XL280 unigenes from five time points from either sense or antisense transcripts, which include 6964 genes predicted to encode proteins. These putative protein-coding genes were further aligned against well-annotated genome sequence of another *C. neoformans* isolate JEC21 to reduce false positives due to variation in gene prediction process (see Materials and methods for details). We identified 2228 protein-coding genes displaying remarkably upregulated expression ($\log_2$|fold-change| > 1.0, q value < 0.01, TPM (transcripts per million mapped)>5) at least at one time point tested (*Supplementary file 1*). These genes were further divided into four groups based on the time point when their transcription was first induced (*Figure 3B*). We reasoned that each group may consist of sets of genes that specifically activate given molecular or differentiation events. Supporting this hypothesis, we found that the four groups individually contained genes responsible for different processes throughout sexual development (filamentation, meiosis and sporulation) (*Figure 3B*) (*Bahn et al., 2005*; *Lin et al., 2005*; *Wang et al., 2012*; *Huang et al., 2015*). A Gene Ontology (GO) analysis using BiNGO program was further performed to systemically explore the biological functions of the genes belonging to the different



**Figure 3.** Gene network specifically orchestrating basidial maturation and meiosis during unisexual mating. (A) Pairwise correlation of normalized TPM between RNA-seq samples obtained at five various time points after unisexual activation (6 hr, 12 hr, 24 hr, 48 hr and 72 hr). Pearson correlation coefficient was calculated using the R package ranges from no correlation (dark blue) to a perfect correlation (red). (B) Line plots show transcriptional induction profiles of genes in each pattern group. For each gene, the normalized expression levels at five time points throughout unisexual development are shown with a pink line (*right*). Pink dots indicate genes with significant induction during mating differentiation (CPM: count per million

*Figure 3 continued on next page*

*Figure 3 continued*

reads, FC: fold change). For each group, the representative genes, which play roles in various phases during sexual reproduction, are indicated (*right*). For each pattern from group II genes (*left*), the average expression levels at five time points across all genes with the pattern are shown with a red line. Tree cluster of group II genes was plotted using Cluster 3.0.

DOI: https://doi.org/10.7554/eLife.38683.010

The following figure supplement is available for figure 3:

**Figure supplement 1.** Group II genes are divided into four sub-groups using tree clustering (Cluster 3.0).

DOI: https://doi.org/10.7554/eLife.38683.011

groups. Genes involved in the response to pheromones, which is determined by the pheromone MAPK pathway, are significantly enriched in group I (*Figure 3B*). In addition, this group also comprised many targets activated by Mat2 and Znf2, which dominate the *Cryptococcus* mating response (early mating process) and filamentous growth (middle mating process), respectively (p<0.001, Fisher's exact test) (*Figures 3B* and *4A*). These findings support the hypothesis that the genes in group I are likely responsible for or related to early or middle mating events. Among the four groups, group II represents the largest gene set and contains 840 genes, which first displayed expression induction at 24 hr after the mating stimulation. Compared with the transcriptomic data from any other time points, the gene expression profile at 24 hr was remarkably different (*Figure 3A*). This finding raises the possibility that this time course likely reflects an important developmental switch, which involves highly organized processes. Indeed, this gene set can be further divided into four sub-groups based on detailed expression signature analysis using tree clustering (*Figure 3B* and *Figure 3—figure supplement 1*). The GO analysis identified various GO terms in these sub-groups and, as expected, revealed enrichment of meiosis genes. In addition, these terms also involve cell wall part (GO: 0044426), fatty acid metabolism (GO: 0006631) and vacuolar protein catabolic process (GO: 007039), which are potentially related to remodeling cell wall or offering energy and metabolic support during basidial maturation. Compared to the group II genes, genes displaying an initial induction at 48 hr (group III) and 72 hr (group IV) are only associated with a few biological processes. These processes mainly include rRNA/protein metabolic processes, which may reflect cellular protein turnover in response to stress after extended inoculation on V8 juice agar, a relatively nutrition-restrictive medium.

Considering the temporal overlap of meiosis and basidial differentiation, we hypothesized that the genes dedicated to the coordination of these two events are likely included in meiosis gene-enriched group II. In this gene group, we specially focused on 'regulatory genes', whose products contain domains associated with DNA or mRNA binding activities, since DNA or mRNA-binding proteins generally function as core regulatory determinants of meiotic progression and fungal cellular differentiation (*van Werven and Amon, 2011*; *Wang and Lin, 2012*). Based on the InterProScan program predication, 19 predicted regulatory genes were identified in gene group II (*Figure 4— source data 1*). We speculated that among these regulatory genes, the ones bridging meiosis and basidial maturation are likely controlled by Pum1 due to its important role in orchestrating these processes. To identify the Pum1-controlled group II regulatory genes, we performed whole-genome RNA-seq profiling at 24 hr after mating induction when group II genes first showed expression induction (*Figure 3B*). It has been previously shown that Pum1 is almost exclusively expressed in hyphae, which constitute only a minor cell population in a mating colony (*Wang et al., 2014*). To avoid noise caused by massive mating cells that do not or poorly express *PUM1*, the $P_{RPL2B}$-*PUM1* strain, which constitutively expresses the *PUM1* gene, was applied to the RNA-seq profiling. The profiling revealed 907 differentially expressed genes (DEGs) in response to the overexpression of Pum1 ($\log_2$|fold-change| > 1.0, q value < 0.01). The number of Pum1 targets explored by the RNA-seq assay was much larger than that identified by the previous transcriptomic assay (85 DEGs) of the *PUM1* overexpression strain based on the microarray analysis, which normally has a lower sensitivity and specificity than RNA-seq technology (*Wang et al., 2014*). Besides, it is notable that the previous microarray experiment was performed at 72 hr post mating stimulation when the expression of many group II genes considerably decreased compared to that at 24 hr (*Figure 3B*). The inappropriate time point used in the previous microarray assay could have led to a failure in comprehensively exploring the group II genes activated by Pum1. Indeed, only 15 genes from group II were found to be upregulated by Pum1 based on the previous transcriptomic assay. By comparison, the current

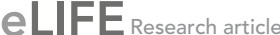

**Figure 4.** Csa1 and Csa2 as the key targets of Pum1 are required for post-meiotic sporulation. (A) Enrichment of genes belonging to different signaling cascades in four gene groups. Among these mating cascades, only the set of genes activated by Pum1 was specifically enriched in group II. Genes used for the enrichment assessment include those encoding the published components (mating MAPK pathway) or the genes activated by the activators (Znf2, Mat2 and Pum1) dominating different sexual stages (*Figure 4—source data 2*). Genes activated by Mat2 and Znf2 ('Mat2-activated

*Figure 4 continued on next page*

*Figure 4 continued*

genes' and 'Znf2-activated genes') are derived from the previous transcriptome data (*Lin et al., 2010*). The gene set 'Pum1-activated genes' is generated based on the RNA-seq analysis of the *PUM1* overexpression strain (*Supplementary file 2*). Only significantly enriched (p<0.001, Fisher's exact test) families are colored. (B) Enriched GO terms of 94 group II genes induced by Pum1 using BiNGO. (C) Dynamic expression of the group II regulators with predicted RNA-binding or DNA-binding domains, whose mRNA levels were induced upon Pum1 overexpression, during unisexual reproduction. (D) Mating phenotypes for wild type XL280 and its isogenic mutant strains. Phenotype scores are represented in distinct colors based on quantitative or semi-quantitative analysis targeting the phenotypes related to sequential differentiation events during unisexual cycle. The results represent experiments from at least three independent mutants. (E) Sporulation phenotypes for XL280 (wild-type) and different Pum1 downstream regulator mutants during unisexual mating. Scale bar: 20 µm.
DOI: https://doi.org/10.7554/eLife.38683.012

The following source data and figure supplements are available for figure 4:

**Source data 1.** Source file for *Figure 4C*.
DOI: https://doi.org/10.7554/eLife.38683.016
**Source data 2.** Source file for *Figure 4A*.
DOI: https://doi.org/10.7554/eLife.38683.017
**Figure supplement 1.** Mating phenotypes of the mutants lacking the regulators downstream of Pum1, including hyphal initiation and hyphal extension.
DOI: https://doi.org/10.7554/eLife.38683.013
**Figure supplement 2.** The absence of Pum1 but not its downstream targets Csa1 and Csa2 adversely affected self-filamentation during unisexual reproduction.
DOI: https://doi.org/10.7554/eLife.38683.014
**Figure supplement 2—source data 1.** Source file for *Figure 1—figure supplement 1*.
DOI: https://doi.org/10.7554/eLife.38683.015

gene expression profiling approach revealed 94 group II genes induced by Pum1, indicating a better sensitivity. Moreover, the group II genes are significantly enriched in Pum1-induced regulon (p=3.95 × 10$^{-7}$, Fisher's exact test) (*Figure 4A*), and many of these genes are related to the meiotic cycle or sexual sporulation (*Figure 4B*), supporting the key role of Pum1 in governing these late sexual events. RNA-seq analysis identified eight group II regulatory genes (*Figure 4C*) that displayed remarkably induced expression in response to Pum1 overexpression. These genes included four potential RNA-binding protein coding genes and four genes predicted to produce DNA-binding protein (*Figure 4C*). We deleted these genes individually, and the impact of the resulting deletion mutants on sequential unisexual differentiation processes (hyphal initiation, hyphal extension and sporulation) were assessed by quantitative or semi-quantitative phenotypic approaches (*Figure 4D*, *Figure 4—figure supplement 1*). We speculated that the regulator involved in the coordination of meiosis and basidial maturation must be important for downstream sporulation but does not affect the earlier differentiation processes, such as filamentation. Our phenotypic assays showed that four of these eight regulators (50%) are involved in post-meiotic sporulation (*Figure 4D*, *Figure 4—figure supplement 1*). Among these four regulators, CNA00260, CNJ00760, and CNB02060 are strictly required for the formation of spores, and we did not observe spore formation in these mutants even after extending the incubation time up to one month. CNA00260, which encodes a ZnF_GATA DNA-binding family protein, exerted a dramatic effect not only on sporulation but also on filamentation, suggesting that it is not specific to *Cryptococcus* fruiting. In contrast, deletion of CNJ00760 or CNB02060 prevented formation of meiospores and led to self-filamentation with an abundance similar to the wild-type level during unisexual reproduction (*Figure 4D*, *Figure 4—figure supplement 1*, *Figure 4—figure supplement 2*). Similarly, these two genes are also critical for bisexual sporulation in bilateral mating assays (*Figure 5A*) and dispensable for bisexual filamentation, which was remarkably defective in bilateral crosses of *pum1Δ* (α *pum1Δ* × a *pum1Δ*) (*Figure 5—figure supplement 1*).

Domain searches using the Motif scan and InterProScan programs revealed that the products of both CNJ00760 and CNB02060 belong to the RRM RNA-binding protein family (*Glisovic et al., 2008*). We named these genes *CSA1* (CNJ00760) and *CSA2* (CNB02060) (*Cryptococcus* sporulation activator). To test whether *CSA1*-activated or *CSA2*-activated sporulation is unique to the XL280 (serotype D) background, *CSA1* and *CSA2* were individually mutated in the JEC21 (serotype D) and H99 (serotype A) backgrounds. Both the serotype D and serotype A strains belong to the *Cryptococcus neoformans* species complex and are considered to have diverged from each other for at least

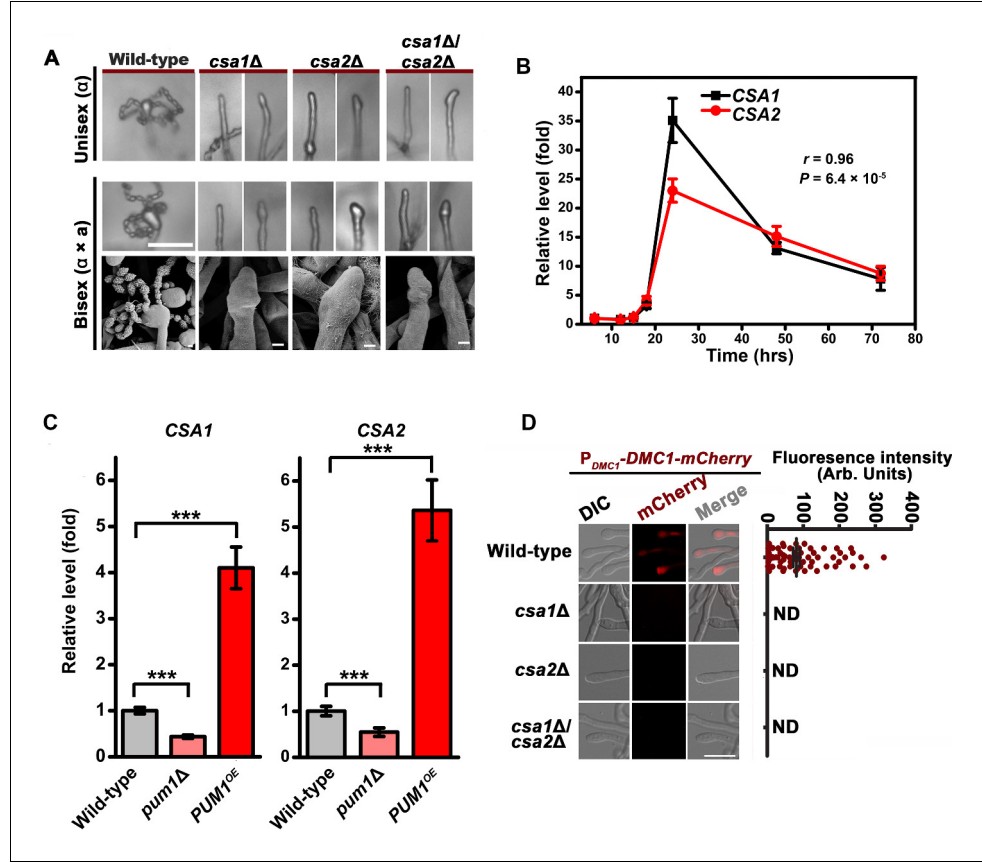

**Figure 5.** Csa1 and Csa2 govern the regulatory coordination of meiosis and basidial differentiation. (**A**) Upper panels indicate sporulation phenotypes for wild-type XL280α, the *csa1Δ* mutant, the *csa2Δ* mutant and the *csa1Δ/ csa2Δ* mutant during unisexual mating; middle and bottom panels illustrate sporulation phenotypes for a wild-type cross between XL280α and XL280a, the *csa1Δ* bilateral mutant cross, the *csa2Δ* bilateral mutant cross, and the *csa1Δ/csa2Δ* bilateral mutant cross. Scale bar: 20 μm (upper and middle panels). Scale bar: 1 μm (bottom panels). (**B**) RT-PCR analysis showed the dynamic expression of *CSA1* and *CSA2* at seven different time points (6 hr, 12 hr, 15 hr, 18 hr, 24 hr, 48 hr and 72 hr) during unisexual mating. Bars show the mean ±SD of six individual experiments. (**C**) RT-PCR analysis indicated that the mRNA levels of both *CSA1* and *CSA2* were positively affected by *PUM1* during unisexual reproduction at 24 hr post inoculation on mating inducing V8 medium. Bars show the mean ± SD of six individual experiments. ***p<0.001, two-tailed Student's *t*-test. (**D**) The images of the fluorescence-labeled strains were taken at 7 days after incubation on V8 medium (*left*). > 50 basidia for each strain were examined for the expression of Dmc1-mCherry (*right*). Scale bar: 10 μm. ND = Not Detected.

DOI: https://doi.org/10.7554/eLife.38683.018

The following source data and figure supplements are available for figure 5:

**Source data 1.** Source file for *Figure 5B,C and D*.

DOI: https://doi.org/10.7554/eLife.38683.021

**Figure supplement 1.** Deletion of *PUM1* but not *CSA1* and *CSA2* attenuated bisexual filamentation in bilateral mating assays.

DOI: https://doi.org/10.7554/eLife.38683.019

**Figure supplement 2.** Deletion of *CSA1* and *CSA2* blocked bisexual sporulation in both laboratory serotype D strain (JEC21) and clinical serotype A isolate (H99).

DOI: https://doi.org/10.7554/eLife.38683.020

18.5 million years (*Xu et al., 2000*). Expectedly, the absence of either of Csa1 or Csa2 in these *Cryptococcus* strains abolished sporulation (*Figure 5A*, *Figure 5—figure supplement 2*). These data demonstrated that the requirement of Csa1 and Csa2 for the formation of meiospores is not limited to XL280 but is conserved among strains in the *C. neoformans* species complex.

## Csa1 and Csa2 are essential for the regulatory coordination of meiosis and basidial differentiation

The RT-PCR analysis of the mRNA levels of *CSA1* and *CSA2* at seven different time points post unisexual induction indicated that their gene expression patterns during unisexual reproduction significantly overlapped ($r = 0.96$, p=$6.4 \times 10^{-5}$, Pearson's test) (*Figure 5B*), which is suggestive of co-regulation. This concept was further supported by the transcriptional evidence showing that the mRNA levels of *CSA1* and *CSA2* were co-upregulated by Pum1 (*Figure 5C*). Next, we asked if defective sporulation in the *csa1Δ* and *csa2Δ* mutants is due to failure in the orchestration of meiosis and basidial maturation, particularly given the important effect of their upstream regulator Pum1 on these two events. To address this question, we first detected the expression of Dmc1-mCherry in the *csa1Δ* and *csa2Δ* mutants during unisexual reproduction. The fluorescence signal was undetected when either of these two genes was absent, suggesting that they are both required for meiotic progression (*Figure 5D*). Early studies have shown that disruption of meiosis-specific genes causes a greatly reduced number of spores or spore chains but cannot completely prevent sporulation, which can be otherwise achieved by the deletion of *CSA1* or *CSA2* (*Figures 4D* and *5A*). This finding highlights the possibility that the blocked sporulation observed in the *csa1Δ* and *csa2Δ* mutants is not only due to defective meiosis but also involves additional mechanism. This idea suggested us to examine whether *CSA1* and *CSA2* affect the formation of the basidium, which offers the physical base underlying the formation of spore chains. The BMS assay indicated that deleting either *CSA1* or *CSA2* dramatically dampened basidial formation and enlargement (maturation) during both unisexual and bisexual reproduction (*Figure 2A*). Basidia, especially mature basidia (BMS >1.6), were reduced to a much lower level in the *csa1Δ* and *csa2Δ* mutants compared with those in the wild-type strain and even the *pum1Δ* mutant (*Figures 2A* and *6A*).

To gain a further insight into the effect of Csa1 and Csa2 on basidial maturation, we investigated the *csa1Δ* and the *csa2Δ* mutants for the patterns of the subcellular localizations of Fad1, which can be used to define the different phases during fruiting body maturation, particularly the early (pattern I and II) and late/mature phases (pattern IV) (*Figure 2C–E*). In the *csa1Δ* mutant,~53.3% and~34.1% of basidia exhibited patterns identical to patterns I and II, respectively, after 7 days incubation on mating-inducing V8 medium. These frequencies were much higher than those detected in the wild-type strain (I =~4.7% and II =~9.7%). Furthermore, up to ~61.9% of basidia in the wild-type XL280α strain achieved the late stage represented by localization pattern IV, while this pattern was detected in only 2.7% of basidia in the absence of Csa1 (*Figure 6B*), suggesting that Csa1 is important for *Cryptococcus* basidial maturation. Despite the dramatic change in the localization features of Fad1 caused by the deletion of *CSA1*, Csa1 cannot control Fad1 expression as revealed by our quantitative fluorescence imaging analysis (*Figure 6—figure supplement 1*). Unlike Csa1, Csa2 contributed to the full expression of Fad1-mCherry, whose abundance was greatly decreased in the mutant lacking *CSA2* (*Figure 6—figure supplement 2A*). Consistently, Csa2 is important for the transcription of *FAD1* during both bisexual and unisexual mating, demonstrating its role as an upstream regulator of *FAD1* (*Figure 6—figure supplement 2B*). Moreover, the *csa2Δ* mutant was nearly devoid of the maturation-represented pattern of Fad1-mCherry (pattern IV) and almost exclusively exhibited early stage patterns (I =~84.3% and II =~13.6%) (*Figure 6B*).

We next performed a detailed phenotypic analysis for evaluating the impact of Csa1 and Csa2 on basidial morphogenesis. Surprisingly, a vast majority of basidia exhibited aberrant morphologies in the absence of Csa1 or Csa2 during unisexual mating (*csa1Δ*:~72.2% and *csa2Δ*:~82.4%) (*Figure 6C*). Most of the irregular basidia in the mutants displayed a 'snake-head'-like phenotype. (*Figure 5A*). In contrast, 'cup-shaped' or 'spindle-shaped' basidia were usually observed in the wild-type XL280 strain in which only ~2.8% of basidia showed morphological abnormalities. Collectively, our findings suggest that Csa1 and Csa2 are critical for basidial formation, maturation and morphogenesis.

## Csa1 can cooperate with Csa2 in the control of basidial maturation

The significance of both Csa1 and Csa2 in the coordination of meiosis and basidial differentiation led us to investigate their genetic interactions. First, we assessed the reciprocal effect of the disruption of one gene on the transcript level of the other. The RT-PCR analysis indicated that the two regulators did not appear to affect the expression of each other, indicating that they may function in parallel (*Figure 6—figure supplement 3*). To further address this hypothesis, we simultaneously

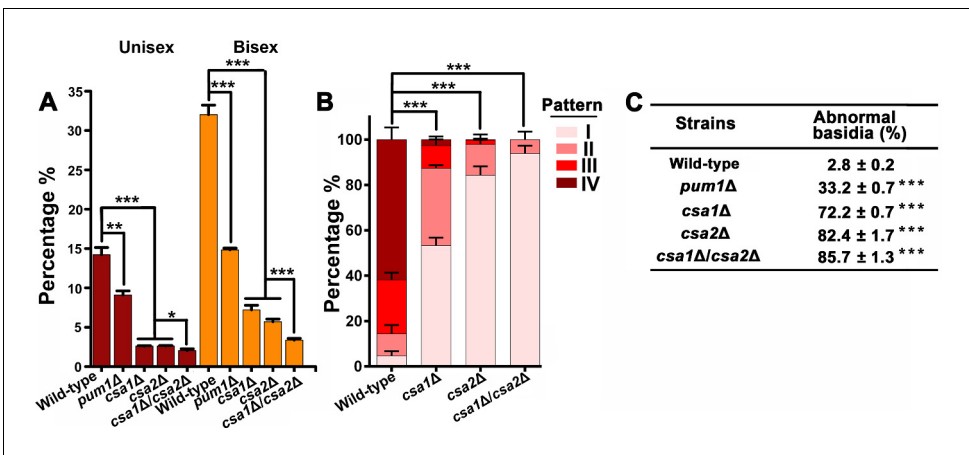

**Figure 6.** Csa1 and Csa2 can function in parallel in basidial maturation and morphogenesis. (A) Compared with either of the single deletion, the *csa1Δ/csa2Δ* mutant displayed a lower number of mature basidia (BMS >1.6) during both unisexual and bisexual reproduction. In both reproduction modes, >150 basidia were examined for each test. Bars show the mean ±SD of three independent experiments. ***p<0.001, *p<0.05, two-tailed Student's-*t* test. (B) Cells were placed onto V8 plate and incubated in the dark at 25°C for the unisexual simulation. >150 basidia for each strain expressing Fad1-mCherry were visualized. Bars show the mean ±SD of three independent experiments. ***p<0.001, two-tailed Student's-*t* test. (C) α cells from XL280 and its derived mutants were incubated on V8 agar in the dark at 25°C to induce the unisexual mating response. Basidia were photographed at 7 days after incubation. >100 basidia for each strain were tested. Data are presented as the mean ± SD from three independent experiments. ***p<0.001 indicates the significant difference compared to the wild-type strain, two-tailed Student's-*t* test.

DOI: https://doi.org/10.7554/eLife.38683.022

The following source data and figure supplements are available for figure 6:

**Source data 1.** Source file for *Figure 6A and B*.

DOI: https://doi.org/10.7554/eLife.38683.029

**Figure supplement 1.** Fad1-mCherry expression was not significantly affected in the absence of Csa1.

DOI: https://doi.org/10.7554/eLife.38683.023

**Figure supplement 1—source data 1.** Source file for *Figure 6—figure supplement 1*.

DOI: https://doi.org/10.7554/eLife.38683.024

**Figure supplement 2.** Fad1 is a downstream target of Csa2.

DOI: https://doi.org/10.7554/eLife.38683.025

**Figure supplement 2—source data 1.** Source file for *Figure 6—figure supplement 2A and B*.

DOI: https://doi.org/10.7554/eLife.38683.026

**Figure supplement 3.** RT-PCR analysis indicated that *CSA1* and *CSA2* do not appear to affect the expression of each other at 48 hr post inoculation on mating inducing V8 medium.

DOI: https://doi.org/10.7554/eLife.38683.027

**Figure supplement 3—source data 1.** Source file for *Figure 6—figure supplement 3*.

DOI: https://doi.org/10.7554/eLife.38683.028

deleted *CSA1* and *CSA2* in the XL280 background. The resulting *csa1Δ/csa2Δ* mutants were applied to compare to the effect of each of the single-deletion mutants on the spatiotemporal expression of meiosis and basidium indicators, as well as basidial maturation and morphogenesis. The expression of Dmc1-mCherry remained undetectable in the *csa1Δ/csa2Δ* mutant (*Figure 5D*). This result was expected because deletion of either genes was sufficient to block the expression of Dmc1 (*Figure 5D*). Of note, compared with either of the single deletion, the double-deletion resulted in a significantly smaller population of large basidia (BMS >1.6), particularly during bisexual mating (*Figures 2A* and *6A*). Consistent with this finding, a modestly higher frequency of irregular basidia was detected in the double-deletion mutant than in either of the single deletion mutants (*Figure 6C*). Furthermore, in the *csa1Δ/csa2Δ* mutant, we failed to observe the mature basidium-specific pattern IV and intermediate pattern (III), which could be detected in the single deletion mutants, although at a much lower level than that in the wild-type strain (*Figure 6B*). In the double-deletion

mutant, Fad1-mCherry exclusively displayed the localization features indistinguishable from the patterns reflecting the early stage of basidial differentiation (I =~93.8% and II =~6.2%). These data suggest that Csa1 can function in concert with Csa2 during fruiting body maturation.

## Discussion

In many microbes, genetically identical cells from a sibling community can be remarkably distinct in cellular shape or physiology (*Mitri and Foster, 2013*; *Wang and Lin, 2015*). Such heterogeneity underlies a source of diversity maximizing microbial survival under numerous environmental and host stresses. For instance, in *C. neoformans*, cells with different morphologies co-exist during the differentiation of the mating community (*McClelland et al., 2004*; *Wang and Lin, 2015*). These morphotypes differ in their tradeoffs related to virulence potential and resistance to given natural or host stress (*Wang and Lin, 2015*). Among these morphotypes, spores, as the final products of mating community differentiation, are considered important infectious propagules due to their resistant nature and size, which is compatible with alveolar deposition (*Sukroongreung et al., 1998*; *Giles et al., 2009*; *Velagapudi et al., 2009*). Considering its importance in terms of *Cryptococcus* biology and infections, important efforts have been undertaken to identify new genes engaged in sporulation (*Bahn et al., 2005*; *Liu et al., 2011*; *Feretzaki and Heitman, 2013*; *Wang et al., 2014*; *Huang et al., 2015*; *Huang and Hull, 2017*). However, much less is known about the regulatory determinant and commitment factor underlying the formation of spores in this important fungal pathogen.

During sexual development, spores are stochastically formed after two parallel events, basidial maturation and meiotic progression. Early studies have reported that sporulation can be perturbed in mutants lacking genes dedicated to meiosis, such as *DMC1* and *SPO11* (*Lin et al., 2005*; *Feretzaki and Heitman, 2013*). This observation suggests that the successful completion of the meiotic cycle is important for sporulation. Notably, mutation of these meiosis-specific genes greatly impairs sporulation but cannot completely abolish it. This finding strongly suggests that *C. neoformans* also involves other commitment mechanism to form spores. The basidium, a hallmark structure of the phylum Basidiomycota that comprises more than 30,000 species, physically supports the formation of spore chains during sexual development (*Kües, 2000*; *Kües and Liu, 2000*; *Wang and Lin, 2011*; *Fu et al., 2015*). Accordingly, basidial maturation may serve as a commitment process for spore production. We showed that meiosis and basidial maturation are coordinated spatiotemporally during both unisex and bisex in *C. neoformans* (*Figure 1E*). Profiling gene induction during mating differentiation further unveiled a special gene group (group II) potentially responsible for the coordination of these two processes (*Figure 3B*). Gene Ontology analysis identified a strong enrichment of cell wall-related genes in this group (*Figure 3B*). This finding probably mirrors the dynamic remodeling of cell wall components during basidial differentiation in *C. neoformans*, particularly given that the re-organization of the cell wall is normally associated with fungal cellular differentiation (*Wang and Lin, 2012*). In addition, multiple genes from group II encode enzymes involved in fatty acid metabolism, including three paralogous genes (CNA05200, CNL05760 and CNF04660) that are predicted to produce peroxisomal/mitochondrial carnitine acetyltransferase (CAT). During fatty acid β-oxidation, CAT is a key enzyme that mediates acetyl-carnitine shuttle to enable the production of energy via the TCA cycle (*Strijbis and Distel, 2010*). An early study has shown that mutants lacking CAT displayed an attenuated production of fruiting body in the cereal fungal pathogen *Gibberella zeae*, suggesting an important role of fatty acid catabolism during sexual structure formation (*Son et al., 2012*). Considering its significance for the production of energy and metabolic intermediates, fatty acid metabolism likely contributes to sustaining fruiting body by providing energy and metabolic supply. Furthermore, the genes associated with lipid/fatty acid metabolism have also been found to be induced during fruiting body differentiation in other basidiomycetes, such as in *Schizophyllum commune* and *Lentinula edodes* (*Ohm et al., 2010*; *Wang et al., 2018*). This may be indicative of the conserved importance of fatty acid metabolism during fruiting body development in divergent fungi.

Despite temporal overlap between meiosis and fruiting body differentiation, the absence of meiosis-specific Dmc1 or Spo11 does not affect basidial initiation and maturation (*Figure 1—figure supplement 1*). This finding demonstrates that basidial differentiation can be achieved independently of meiotic progression and that a shared regulatory program may be responsible for the coordination

of these events (*Figure 7*). This hypothesis was confirmed by the identification of the regulatory circuitry formed by Pum1, Csa1 and Csa2, which are the targets of Pum1 (*Figure 7*). Compared with their upstream regulator, Csa1 and Csa2 are more specific and essential for directing the co-regulation of meiosis and basidial differentiation (*Figure 4E*, *Figure 4—figure supplement 1*, *Figure 4—figure supplement 2*, *Figure 5A* and *Figure 5—figure supplement 1*). The domain prediction analysis indicated that both Csa1 and Csa2 belong to the RRM protein family. In addition to Csa1 and Csa2, many RRM family members in different eukaryotic kingdoms have been reported to control sexual development or meiosis, but most of them are not similar in their protein sequences, except for Mei2 and its homologs (*Jeffares et al., 2004*). Mei2 has been demonstrated to be the master regulator of meiosis in *Schizosaccharomyces pombe*, and the genes encoding its orthologs were found in several groups of eukaryotes (*Jeffares et al., 2004*). Thus, *MEI2-like* genes are considered to have arisen early in the eukaryotic evolution. Based on blast analysis, Csa2, but not Csa1, displays significant similarity with Mei2 in the C-terminal RRM motif. The functions conducted by Csa2 and its ortholog in *S. pombe* during sexual differentiation are not identical, although both are required for meiosis. Csa2 governs basidium formation in *C. neoformans* (*Figures 2A* and *5D*). In contrast, Mei2 appears not to be required for the formation of ascus (*Nakayama et al., 1985*), which is the sexual structure of *S. pombe* analogous to basidium in *C. neoformans*, suggesting a divergent evolution. Consistently, accumulating studies on Mei2-like proteins in plants have demonstrated that their functions are not limited to meiosis but also associated with other biological processes, such as leaf development and vegetative growth (*Kaur et al., 2006*; *Kawakatsu et al., 2006*). The functional divergence is very common among orthologs of mRNA binding proteins, and is likely achieved through altered downstream targets or interaction partners (*Hogan et al., 2015*). During meiosis in fission yeast, Mei2 binds the noncoding RNA meiRNA and sequesters Mmi1, an inhibitor of meiosis, through Mei2-Mmi1 interactions (*van Werven and Amon, 2011*). However, there is no gene from *C. neoformans* genome showing significant homology to the ones in *S. pombe* that produce meiRNA or Mmi1. The divergence of targets controlled by Mei2 and Csa2 is probably attributed to considerable differences between their sequences beyond the C-terminal RRM motif. Consistent with this notion, the region 429–532 upstream of the C-terminal RRM contains two residues (Ser 438 and Thr 527) for phosphorylation by the kinase Pat1 and is essential for the function of Mei2 (*Watanabe et al., 1997*), but this region is missing from Csa2 protein. Blast analysis indicated that Csa2 orthologs from different basidiomycetes share a high similarity in the full protein sequence (greater than 50% coverage). Intriguingly, most fungi harboring Csa2-like protein coding genes also have genes producing Csa1 homologs (greater than 30% identity and greater than 50% coverage). Phylogenetic analysis demonstrated that the fungal species containing both *CSA1-like* and *CSA2-like*

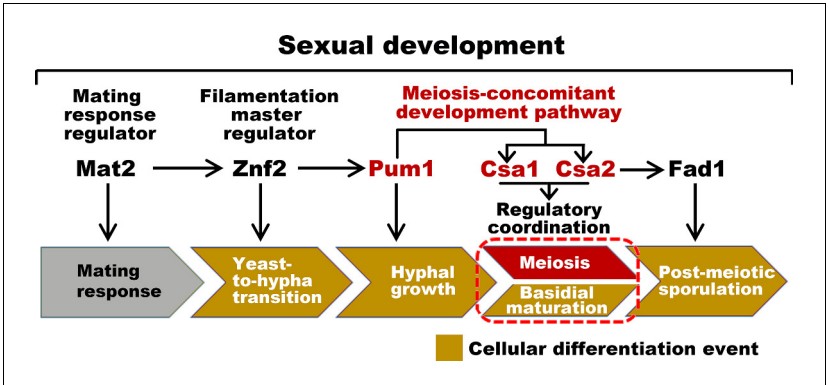

**Figure 7.** Sexual control in *C.neoformans*. Model describing the genes responsible for sequential events during sexual reproduction. Csa1 and Csa2 governs the regulatory coordination of basidial maturation and meiosis, which is required for sporulation.

DOI: https://doi.org/10.7554/eLife.38683.030

The following figure supplement is available for figure 7:

**Figure supplement 1.** Phylogenetic tree of Csa1 orthologs.

DOI: https://doi.org/10.7554/eLife.38683.031

genes belong to the Tremellales clade (*Figure 7—figure supplement 1*). This may suggest the conserved and concerted function of these homologs in coordinating basidial maturation and meiotic progression in Tremellales.

## Materials and methods

**Key resources table**

| Reagent type (species) or resource | Designation | Source or reference | Identifiers | Additional information |
|---|---|---|---|---|
| Genetic reagent (*C. neoformans* species complex) | XL280, *MAT*α, wild-type | PMID: 17112316 | | |
| Genetic reagent (*C. neoformans* species complex) | XL280, *MAT*a, wild-type | PMID: 23670559 | | |
| Genetic reagent (*C. neoformans* species complex) | JEC21, *MAT*α, wild-type | PMID: 10512666 | | |
| Genetic reagent (*C. neoformans* species complex) | JEC20, *MAT*a, wild-type | PMID: 10512666 | | |
| Genetic reagent (*C. neoformans* species complex) | KN99, *MAT*α, wild-type | PMID: 12933823 | | |
| Genetic reagent (*C. neoformans* species complex) | KN99, *MAT*a, wild-type | PMID: 12933823 | | |
| Genetic reagent (*C. neoformans* species complex) | XL280, *MAT*α, *CSA2::NEO*$^r$ | This study | | See Materials and methods, 'Gene disruption and overexpression' |
| Genetic reagent (*C. neoformans* species complex) | XL280, *MAT*a, *CSA2::NEO*$^r$ | This study | | |
| Genetic reagent (*C. neoformans* species complex) | H99, *MAT*α, *CSA2::NEO*$^r$ | This study | | |
| Genetic reagent (*C. neoformans* species complex) | H99, *MAT*a, *CSA2::NEO*$^r$ | This study | | |
| Genetic reagent (*C. neoformans* species complex) | JEC21, *MAT*α, *CSA2::NEO*$^r$ | This study | | |
| Genetic reagent (*C. neoformans* species complex) | JEC20, *MAT*a, *CSA2::NEO*$^r$ | This study | | |
| Genetic reagent *C. neoformans* species complex | XL280, *MAT*α, *FAD1::NEO*$^r$ | PMID: 24901238 | | |
| Genetic reagent (*C. neoformans* species complex) | XL280, *MAT*a, *FAD1::NEO*$^r$ | This study | | |
| Genetic reagent (*C. neoformans* species complex) | XL280, *MAT*α, *PUM1::NEO*$^r$ | PMID: 24901238 | | |
| Genetic reagent (*C. neoformans* species complex) | XL280, *MAT*a, *PUM1::NEO*$^r$ | PMID: 24901238 | | |
| Genetic reagent (*C. neoformans* species complex) | XL280, *MAT*α, *CSA1::NAT*$^r$ | This study | | |

*Continued on next page*

*Continued*

| Reagent type (species) or resource | Designation | Source or reference | Identifiers | Additional information |
|---|---|---|---|---|
| Genetic reagent (*C. neoformans* species complex) | XL280, *MAT*a, *CSA1::NAT*r | This study | | |
| Genetic reagent (*C. neoformans* species complex) | H99, *MATα, CSA1::NEO*r | This study | | |
| Genetic reagent (*C. neoformans* species complex) | H99, *MAT*a, *CSA1::NEO*r | This study | | |
| Genetic reagent (*C. neoformans* species complex) | JEC21, *MATα, CSA1::NEO*r | This study | | |
| gGenetic reagent (*C. neoformans* species complex) | JEC20, *MAT*a, *CSA1::NEO*r | This study | | |
| Genetic reagent (*C. neoformans* species complex) | XL280, *MATα, DMC1::NEO*r | This study | | |
| Genetic reagent (*C. neoformans* species complex) | XL280, *MATα, SPO11::NEO*r | PMID: 23966871 | | |
| Genetic reagent (*C. neoformans* species complex) | XL280, *MATα, CSA2::NEO*r, *CSA1::NAT*r | This study | | |
| Genetic reagent (*C. neoformans* species complex) | XL280, *MAT*a, *CSA2::NEO*r, *CSA1::NAT*r | This study | | |
| Genetic reagent (*C. neoformans* species complex) | XL280, *MATα*, $P_{RPBL2B}$-*PUM1-HYG* | This study | | |
| Genetic reagent (*C. neoformans* species complex) | XL280, *MATα*, $P_{DMC1}$-*DMC1-m Cherry-3'UTR-HYG* | PMID: 24901238 | | |
| Genetic reagent (*C. neoformans* species complex) | XL280, *MAT*a, $P_{DMC1}$-*DMC1-mCherry-3'UTR-HYG* | This study | | |
| Genetic reagent (*C. neoformans* species complex) | XL280, *MATα*, *CSA2::NEO*r, $P_{DMC1}$-*DMC1-mCherry-3'UTR-HYG* | This study | | |
| Genetic reagent (*C. neoformans* species complex) | XL280, *MATα*, *CSA1::NAT*r, $P_{DMC1}$-*DMC1-mCherry-3'UTR-HYG* | This study | | |
| Genetic reagent (*C. neoformans* species complex) | XL280, *MATα*, $P_{FAD1}$-*FAD1-mCherry-HYG* | This study | | |
| Genetic reagent (*C. neoformans* species complex) | XL280, *MATα*, *CSA2::NEO*r, $P_{FAD1}$-*FAD1-mCherry-HYG* | This study | | |
| Genetic reagent (*C. neoformans* species complex) | XL280, *MATα*, $P_{FAS1}$-*FAS1-mCherry-HYG* | This study | | |
| Genetic reagent (*C. neoformans* species complex) | XL280, *MATα*, $P_{DHA1}$-*DHA1-mCherry-HYG* | This study | | |

*Continued on next page*

*Continued*

| Reagent type (species) or resource | Designation | Source or reference | Identifiers | Additional information |
|---|---|---|---|---|
| Genetic reagent (*C. neoformans* species complex) | XL280, *MATα*, CSA1::NAT^r, P_{FAD1}-FAD1-mCherry-HYG | This study | | |
| Genetic reagent (*C. neoformans* species complex) | XL280, *MATα*, CSA1::NAT^r, P_{FAD1}-FAD1-mCherry-HYG | This study | | |
| Genetic reagent (*C. neoformans* species complex) | XL280, *MATα*, CSA1::NAT^r, CSA2::NEO^r, P_{DMC1}-DMC1-mCherry-3'UTR-HYG | This study | | |
| Genetic reagent (*C. neoformans* species complex) | XL280, *MATα*, CSA1::NAT^r, CSA2::NEO^r, P_{FAD1}-FAD1-mCherry-3'UTR-HYG | This study | | |
| Genetic reagent (*C. neoformans* species complex) | XL280, *MATα*, CNB02310::NEO^r | This study | | |
| Genetic reagent (*C. neoformans* species complex) | XL280, *MATα*, CNF03810::NEO^r | This study | | |
| Genetic reagent (*C. neoformans* species complex) | XL280, *MATα*, CNF01980::NEO^r | This study | | |
| Genetic reagent (*C. neoformans* species complex) | XL280, *MATα*, CNB05180::NEO^r | This study | | |
| Genetic reagent (*C. neoformans* species complex) | XL280, *MATα*, CNF00260::NEO^r | This study | | |
| Genetic reagent (*C. neoformans* species complex) | XL280, *MATα*, CNG01790::NEO^r | This study | | |
| Software, algorithm | RStudio Version 1.1.456 | RStudio | RRID:SCR_000432 | |
| Software, algorithm | FastQC v0.11.5 | | RRID:SCR_014583 | |
| Software, algorithm | DEseq2 v1.16.1 | | RRID:SCR_016533 | |
| Software, algorithm | Hisat2 v2.1.0 | | RRID:SCR_015530 | |
| Software, algorithm | BiNGO v3.0.3 | | RRID:SCR_005736 | |
| Software, algorithm | Graphpad Prism 6 | Graphpad | RRID:SCR_002798 | |
| Sequence-based reagent | Wanglab959 (knockout primer pairs) | This study | | TTGTCACCAACCTATCCGCTAC |
| Sequence-based reagent | Wanglab960 (knockout primer pairs) | This study | | CAGTTTGCTCTTATTCCCACTCC |
| Sequence-based reagent | Wanglab961 (knockout primer pairs) | This study | | CTGGCCGTCGTTTTACGAAG CACTTGGTGAATGAGACATT |
| Sequence-based reagent | Wanglab962 (knockout primer pairs) | This study | | GTCATAGCTGTTTCCTGCA CCGCCCTTACGATTATACATCT |
| Sequence-based reagent | Wanglab963 (knockout primer pairs) | This study | | GTTGTGAGTGTCATGAGTGTCATTG |

*Continued on next page*

*Continued*

| Reagent type (species) or resource | Designation | Source or reference | Identifiers | Additional information |
|---|---|---|---|---|
| Sequence-based reagent | Wanglab964 (knockout primer pairs) | This study | | CCTCTTCTGCCAATAACCCTTTT |
| Sequence-based reagent | Wanglab2195 (knockout primer pairs) | This study | | CATCCCCAGAACACGCTGAT |
| Sequence-based reagent | Wanglab2196 (knockout primer pairs) | This study | | TCCGGCCATTAAGATCCGTG |
| Sequence-based reagent | Wanglab2197 (knockout primer pairs) | This study | | AAAACGGCAACAGTCAAGGC |
| Sequence-based reagent | Wanglab2198 (knockout primer pairs) | This study | | CTGGCCGTCGTTTTACGTTGT TAAAGGCAGTTGAGCGA |
| Sequence-based reagent | Wanglab2199 (knockout primer pairs) | This study | | GTCATAGCTGTTTCCTGAAGG CATCACTTCGTTTGGC |
| Sequence-based reagent | Wanglab2200 (knockout primer pairs) | This study | | AACCATAGGATGTGCCACGC |
| Sequence-based reagent | Wanglab953 (knockout primer pairs) | This study | | CCGTAGGCTTATCCCAGTCAGA |
| Sequence-based reagent | Wanglab954 (knockout primer pairs) | This study | | GTGGAAGGCAAGAGTTGGTGTT |
| Sequence-based reagent | Wanglab955 (knockout primer pairs) | This study | | CTGGCCGTCGTTTTACACAT TTCCAGAAGAGGCAAGAAGA |
| Sequence-based reagent | Wanglab956 (knockout primer pairs) | This study | | GTCATAGCTGTTTCCTGGGG TAGAAGAACGTCAAACAACTAA |
| Sequence-based reagent | Wanglab957 (knockout primer pairs) | This study | | CCTTGGCAACAGTAGGCTTCTG |
| Sequence-based reagent | Wanglab958 (knockout primer pairs) | This study | | GGAAGGGAGTGGTGAGGTTGAA |
| Sequence-based reagent | Wanglab2461 (knockout primer pairs) | This study | | GGGCCTGAAAAGTATGAAGTCC |
| Sequence-based reagent | Wanglab2462 (knockout primer pairs) | This study | | TAGCCTTTCCACCACAGCAGC |
| Sequence-based reagent | Wanglab2463 (knockout primer pairs) | This study | | CTGGCCGTCGTTTTACGTAG CGGTTTCGACGGACATAT |
| Sequence-based reagent | Wanglab2464 (knockout primer pairs) | This study | | GTCATAGCTGTTTCCTGGGA AGAGGAGGAGACCAAGGAG |
| Sequence-based reagent | Wanglab2465 (knockout primer pairs) | This study | | ATCCTTTGTCCAACCCGTGAG |
| Sequence-based reagent | Wanglab2466 (knockout primer pairs) | This study | | GCCCATGTCGCATTACGTAAAG |
| Sequence-based reagent | Wanglab2423 (knockout primer pairs) | This study | | AGCCATTCGGCTCTTATCGC |
| Sequence-based reagent | Wanglab2424 (knockout primer pairs) | This study | | AGCGACTGCGACCATTATGT |
| Sequence-based reagent | Wanglab2425 (knockout primer pairs) | This study | | CTGGCCGTCGTTTTACATGG AGGCGTTGGAGAATCC |
| Sequence-based reagent | Wanglab2426 (knockout primer pairs) | This study | | GTCATAGCTGTTTCCTGGCA AGACGTGCATACCCTCTA |
| Sequence-based reagent | Wanglab2427 (knockout primer pairs) | This study | | GCTTCAGTATGCCAACCCCT |
| Sequence-based reagent | Wanglab2428 (knockout primer pairs) | This study | | CGAGAGAAGGGAAAGCGAGG |
| Sequence-based reagent | Wanglab2201 (knockout primer pairs) | This study | | GGAGAGATCAGAGGCAGCAC |

*Continued on next page*

*Continued*

| Reagent type (species) or resource | Designation | Source or reference | Identifiers | Additional information |
|---|---|---|---|---|
| Sequence-based reagent | Wanglab2202 (knockout primer pairs) | This study | | CGTCGTGGAAAAGGTGAGGA |
| Sequence-based reagent | Wanglab2203 (knockout primer pairs) | This study | | TCCGGATTTCTCAAGTGGGC |
| Sequence-based reagent | Wanglab2204 (knockout primer pairs) | This study | | CTGGCCGTCGTTTTACGCT CTAGCATTTGCGGGGAT |
| Sequence-based reagent | Wanglab2205 (knockout primer pairs) | This study | | GTCATAGCTGTTTCCTGTGACT CCCCCTCCAGAAAGC |
| Sequence-based reagent | Wanglab2206 (knockout primer pairs) | This study | | AACCAAAATGGCTCCGGACA |
| Sequence-based reagent | Wanglab2682 (knockout primer pairs) | This study | | TTGCAACCATCCGAGGTCAA |
| Sequence-based reagent | Wanglab2683 (knockout primer pairs) | This study | | GAAATCCGACACCTCCCTGG |
| Sequence-based reagent | Wanglab2684 (knockout primer pairs) | This study | | CTGGCCGTCGTTTTACGG GATGTTTGTCCCTTTCGC |
| Sequence-based reagent | Wanglab2685 (knockout primer pairs) | This study | | GTCATAGCTGTTTCCTGAC CAGTAAGAAGCGGTGACA |
| Sequence-based reagent | Wanglab2686 (knockout primer pairs) | This study | | AGCGCTCGACTAGCTTTCTC |
| Sequence-based reagent | Wanglab2687 (knockout primer pairs) | This study | | GGATCCAAGACCTCCGATGG |
| Sequence-based reagent | Wanglab3060 (knockout primer pairs) | This study | | AGCGATAAGCCAGCAAGAGTT |
| Sequence-based reagent | Wanglab3061 (knockout primer pairs) | This study | | CCTCGAACCCGATACTGACG |
| Sequence-based reagent | Wanglab3062 (knockout primer pairs) | This study | | AGCTTAGAATAGCGACCGCC |
| Sequence-based reagent | Wanglab3063 (knockout primer pairs) | This study | | CTGGCCGTCGTTTTACTGTGA GAGTCGGCTGATAGGA |
| Sequence-based reagent | Wanglab3064 (knockout primer pairs) | This study | | GTCATAGCTGTTTCCTGGTGGA ACCTAATTGCACCGC |
| Sequence-based reagent | Wanglab3065 (knockout primer pairs) | This study | | ATGGCGAGTTGCTTTCATGC |
| Sequence-based reagent | Wanglab3066 (knockout primer pairs) | This study | | TAATGTCGCTGAAGGGCCTG |
| Sequence-based reagent | Wanglab3067 (knockout primer pairs) | This study | | CCAAGGGTCAGCTATCCAGC |
| Sequence-based reagent | Wanglab3068 (knockout primer pairs) | This study | | CCGTAACCGGTGAGACATCA |
| Sequence-based reagent | Wanglab3069 (knockout primer pairs) | This study | | CTGGCCGTCGTTTTACGAGA CGAATGAGCTGTGGCA |
| Sequence-based reagent | Wanglab3070 (knockout primer pairs) | This study | | GTCATAGCTGTTTCCTGTCAA GTCATGCCTGTGATCCT |
| Sequence-based reagent | Wanglab3071 (knockout primer pairs) | This study | | AGATCCTGGAGGGAACGGAT |
| Sequence-based reagent | Wanglab3072 (knockout primer pairs) | This study | | TTAGCTCGCCCTCGCTTATT |
| Sequence-based reagent | Wanglab3073 (knockout primer pairs) | This study | | AGCCAACCCATTTACCGACT |
| Sequence-based reagent | Wanglab3074 (knockout primer pairs) | This study | | CGTTGGACAATGGAGTGAGGA |

*Continued*

| Reagent type (species) or resource | Designation | Source or reference | Identifiers | Additional information |
|---|---|---|---|---|
| Sequence-based reagent | Wanglab3075 (knockout primer pairs) | This study | | CTGGCCGTCGTTTTACGGGGA TGAAGGGAGCTAAAGG |
| Sequence-based reagent | Wanglab3076 (knockout primer pairs) | This study | | GTCATAGCTGTTTCCTGGAAG CCTTTGCATTTGACCCT |
| Sequence-based reagent | Wanglab3077 (knockout primer pairs) | This study | | GGACAGAGGCCGTCAACATA |
| Sequence-based reagent | Wanglab3646 (knockout primer pairs) | This study | | CTAACGACAACAAGAAACCACGAC |
| Sequence-based reagent | Wanglab3647 (knockout primer pairs) | This study | | CTGGCCGTCGTTTTACAGGCGGA GGAAGGTAGGAGAA |
| Sequence-based reagent | Wanglab3648 (knockout primer pairs) | This study | | GTCATAGCTGTTTCCTGGTAGGTAA TGTTGACGGTGGTGA |
| Sequence-based reagent | Wanglab3649 (knockout primer pairs) | This study | | GTCTTAGTGGTCTGAGCCGAATAC |
| Sequence-based reagent | Wanglab3650 (knockout primer pairs) | This study | | AGGACGCTATTCGCTCTATCGG |
| Sequence-based reagent | Wanglab3651 (knockout primer pairs) | This study | | GATCCTTCACCCTGACTCTGTTCA |
| Sequence-based reagent | Wanglab3261 (knockout primer pairs) | This study | | ACTCATGCCTACCCATTGCC |
| Sequence-based reagent | Wanglab3262 (knockout primer pairs) | This study | | GCGACTCACTGAGCTTGACA |
| Sequence-based reagent | Wanglab3263 (knockout primer pairs) | This study | | CGGGCTTTACACCTACTCGG |
| Sequence-based reagent | Wanglab3264 (knockout primer pairs) | This study | | CTGGCCGTCGTTTTACTCTGC TTGTACGTCAGCGAT |
| Sequence-based reagent | Wanglab3265 (knockout primer pairs) | This study | | GTCATAGCTGTTTCCTGAGTGA AGAGACTTGACGCTCG |
| Sequence-based reagent | Wanglab3266 (knockout primer pairs) | This study | | ACTAGCCCGAAGTGATGGGA |
| Sequence-based reagent | Wanglab3267 (knockout primer pairs) | This study | | GGCGCGTTGTAAAGCAGTAG |
| Sequence-based reagent | Wanglab3268 (knockout primer pairs) | This study | | TCTCCCCTCGGAAACAGCTA |
| Sequence-based reagent | Wanglab3269 (knockout primer pairs) | This study | | AGCACCTTTGCGATGTCTGA |
| Sequence-based reagent | Wanglab3270 (knockout primer pairs) | This study | | CTGGCCGTCGTTTTACGTTC CTGGACCCTTGATCCC |
| Sequence-based reagent | Wanglab3271 (knockout primer pairs) | This study | | GTCATAGCTGTTTCCTGGC AGTAACGGTCCTGTTCCA |
| Sequence-based reagent | Wanglab3272 (knockout primer pairs) | This study | | GTTCGATCAGAAACACGGCG |
| Sequence-based reagent | Wanglab857 (qRT-PCR primer) | This study | | CGTCACCACTGAAGTCAAGT |
| Sequence-based reagent | Wanglab858 (qRT-PCR primer) | This study | | AGAAGCAGCCTCCATAGG |
| Sequence-based reagent | Wanglab3401 (qRT-PCR primer) | This study | | AGACTCGACCACAGGCAG |
| Sequence-based reagent | Wanglab3402 (qRT-PCR primer) | This study | | AAAGGACAGGGTCAGGGTT |
| Sequence-based reagent | Wanglab2583 (qRT-PCR primer) | This study | | TTCTGCCGTAATGGGTGTCA |

*Continued*

| Reagent type (species) or resource | Designation | Source or reference | Identifiers | Additional information |
|---|---|---|---|---|
| Sequence-based reagent | Wanglab2584 (qRT-PCR primer) | This study | | TCGTAAGGGCGGTGTTGTG |
| Sequence-based reagent | Wanglab2585 (qRT-PCR primer) | This study | | GTGAGATTATTGCCCGTGATGA |
| Sequence-based reagent | Wanglab2586 (qRT-PCR primer) | This study | | TTGGAGACGCCAGGGATGT |
| Sequence-based reagent | Wanglab855 | This study | | CTCTGGTTGGCACGGTG |
| Sequence-based reagent | Wanglab856 | This study | | CGTCGGTCAATCTTCTCG |
| Sequence-based reagent | Wanglab2689 (overexpression primer) | This study | | TTTGCATTGCGGCCGCAGGG GTGAATCGATATTCGACGC |
| Sequence-based reagent | Wanglab2690 (overexpression primer) | This study | | GGATAATTGCGATCGCCAGCTG GAGAGTGACAGACTTGG |

## Strains and growth conditions

The strains used in this study are listed in the Key Resources Table. *Cryptococcus* yeast cells were cultured on YPD solid medium (1% yeast extract, 2% Bacto peptone, 2% dextrose, and 2% Bacto agar) at 30°C for routine growth. Unisexual and bisexual mating assays were carried out on V8 solid medium (0.5 g/liter $KH_2PO_4$, 4% Bacto agar, and 5% V8 juice) in the dark at 25°C (V8 pH seven agar for serotype D strains and V8 pH five agar for serotype A H99 strain). YPD media containing nourseothricin (*NAT*), G418 (*NEO*) or hygromycin (*HYG*) were used for selecting the *Cryptococcus* transformants generated by electroporation and biolistic transformation.

## Filamentation, sporulation assays and BMS assay

For bisexual filamentation and sporulation assays, congenic α and a cells (XL280α/a, JEC21α/a and H99α/**a**) were cultured on YPD medium separately overnight at 30°C. Cells were then collected by centrifugation. Equal numbers ($OD_{600}$ = 1.0) of collected cells from opposite mating types were co-incubated on V8 medium in the dark at 25°C for mating stimulation. For self-filamentation and unisexual sporulation assays (serotype D XL280α strain background), the cells were spotted on V8 medium alone. Both bisexual and unisexual mating phenotypes were examined microscopically for production of mating hyphae and chains of basidiospores.

For the BMS analysis, α cells alone (unisexual mating) or α-a cell mixtures (α-a bisexual mating) were cultured on V8 medium in the dark at 25°C to stimulate mating. In most BMS assays performed in this study, the cells were harvested at 7 days post mating stimulation, unless otherwise indicated. In both reproduction modes, the cells displayed evident heterogeneity in morphotypes (mostly yeast and hyphae), and ample hyphae are normally formed on the edge of the mating colony, especially during unisexual mating. Regardless of their morphotypes, the cells were entirely scraped off the edge of mating patches to avoid bias. All cells were harvested, vortexed and suspended in 20 μl fixative (3.7% formaldehyde and 1% Triton X-100 in PBS buffer). Mating cells (2 μl) were subsequently dropped onto a glass slide for microscopic examination. Among the cells, most hyphae tended to form 'hyphal clusters' due to cell aggregation. Over 100 hyphae with or without basidia from different 'clusters' in each sample were randomly chosen for the BMS calculation. A BMS of 1.0 was arbitrarily set as the threshold to define the basidial state. In the BMS assays of sporulated basidia that constitute a minority of the basidial population, at least 50 basidia were examined for each sample. The diameters of basidia and their connected hyphae were measured using a Zeiss Imager A2-M2 imaging system with AxioCam MRm camera software Zen 2011 (Carl Zeiss Microscopy).

## Gene disruption and overexpression

For the gene disruption, overlapping PCR products were generated with a *NEO* or NAT resistance cassette and 5′ and 3′ flanking sequences (1.0 ~ 1.5 kb) of the coding regions of selected genes from *Cryptococcus* strains as we previously described (*Wang et al., 2012*; *Wang et al., 2013*). The

PCR products were introduced into the *Cryptococcus* strains *via* biolistic transformation. The resulting mutants, in which the genes were replaced by homologous recombination, were confirmed by PCR. For Pum1 overexpression, *PUM1* gene open reading frame were amplified by PCR, and the amplified fragments were digested with FseI and PacI. The digested fragment was then introduced into the copper-inducible plasmid pXC (*Wang et al., 2013*). The plasmid was digested with Not1 and FseI to remove the copper-inducible promoter ($P_{CTR4}$), which was replaced by the promoter region of *RPL2B* by ligation to generate the $P_{RPL2B}$-*PUM1* overexpression system. The primers used for the gene disruption and overexpression are listed in the Key Resources Table.

## Microscopy and fluorescence

The mCherry protein was fused to the C terminus of Dmc1, Fad1, Fas1, and Dha1. The coding regions of the mCherry-fused products were placed under the control of their native promoters. The constructs were introduced into *Cryptococcus* cells using electroporation (*Wang et al., 2012*). The strains LL174α ($P_{DMC1}$-*DMC1*-mCherry) and LL168α ($P_{FAD1}$-*FAD1*-mCherry) were subsequently used as the parental strains to generate the isogenic mutant strains (HG516α, LL178α, HG519α and LL194α), in which selected genes were knocked out. The strains used in this study are listed in the Key Resources Table. To examine the protein subcellular localization, the cells were placed onto glass slide and visualized by a Zeiss Axioplan two imaging system with AxioCam MRm camera software Zen 2011 (Carl Zeiss Microscopy).

## Scanning electron microscopy (SEM)

SEM was performed with the assistance of a Beijing Regional Center of Life Science Instrument, Chinese Academy of Sciences. The samples were prepared for SEM as previously described (*Fu and Heitman, 2017*). For all SEM assays performed in this study, the cell samples were obtained from bilateral mating. The α-**a** mixtures were cultivated on V8 solid medium at 25°C for 7 days in the dark. The colonies were excised and fixed in phosphate-buffered glutaraldehyde (pH 7.2) at 4°C for more than 2 hr. Samples were then rinsed with ddH$_2$O three times for 6 min, 7 min and 8 min, respectively. The rinsed samples were dehydrated through a graded ethanol series (50%, 70%, 85% and 95%) for 14 min for each concentration and then 100% ethanol three times for 15 min. After dehydration, the cells were critical-point dried with liquid CO$_2$ (Leica EM CPD300, Germany) and sputter coated with gold-palladium (E-1045 ion sputter, Hitachi, Japan). The samples were viewed under a Quanta200 scanning electron microscope (FEI, America).

## RNA purification and qPCR analyses

For the RNA-seq analysis, the wild-type XL280 strain and isogenic Pum1 overexpression mutants were cultured in YPD liquid medium (extremely mating-suppressing condition) at 30°C overnight. The overnight culture was then washed with cold water and dropped on V8 agar (pH = 7) for mating induction. The cells were collected at different time points post mating stimulation for the isolation of total RNA. The total RNA was extracted using TRIzol Reagent (CW0580M, CWBIO) and an Ultrapure RNA Kit (CW0581M, CWBIO) according to the manufacturer's instructions. Total RNA (2 μg) was subjected to gDNase treatment, and single-stranded cDNA was synthesized by a Fastquant RT Kit (with gDNase, KR106-02, Tiangen) according to the manufacturer's instructions. The relative mRNA level of selected genes was measured by real time RT-PCR using RealMaster Mix (SYBR Green, FP202-02, TIANGEN) in a CFX96 TouchTM Real-time PCR detection system (Bio-Rad). The primers used for qPCR in this study are listed in the Key Resources Table. The relative transcript levels were normalized to those of the reference housekeeping gene *TEF1* and determined using the $2^{-\Delta\Delta CT}$ approach.

## RNA-seq and data analysis

The total RNA from each sample was purified as previously described (*Wang et al., 2012*). RNA purity was assessed using a Nano Photometer spectrophotometer (IMPLEN, CA, USA), and the RNA concentration was measured using Qubit RNA Assay Kit in a Qubit 2.0 Fluorometer (Life Technologies, CA, USA). The RNA integrity was assessed using the RNA Nano 6000 Assay Kit of a Bioanalyzer 2100 system (Agilent Technologies, CA, USA). The transcriptome library for sequencing was generated using a VAHTSTM Stranded mRNA-seq v2 Library Prep Kit for Illumina (Vazyme Biotech Co.,

Ltd, Nanjing, China) following the manufacturer's recommendations. The clustering of the index-coded samples was performed using VAHTS RNA Adapters set1/set2 for Illumina (Vazyme Biotech Co., Ltd, Nanjing, China) according to the manufacturer's instructions. After clustering, the libraries were sequenced on an Illumina platform.

The raw images were transformed into raw reads by base calling using CASAVA (http://www.illumina.com/support/documentation.ilmn). Then, the raw reads in a fastq format were first processed using in-house Perl scripts. Clean reads were obtained by removing the reads with adapters, such as the reads in which unknown bases exceeded 5%. The low-quality reads were defined by a low-quality base, and the sequencing quality score was no more than 10. Additionally, the Q20, Q30, and GC contents of the clean data were calculated.

The quality of sequenced clean data was assessed using FastQC v0.11.5 software. Then,~2 GB clean data for each sample (representing over 100 x coverage) were mapped to the genome sequence of *C. neoformans* XL280 (XL280α) using Hisat2 v2.1.0. The gene expression level was measured in TPM by Stingtie v1.3.3 to determine unigenes. All unigenes were subsequently aligned against the well-annotated genome of JEC21α (which is congenic to JEC20**a** that served as the parent strain to generate XL280α through a cross with B3501α). The protein coding genes found in both genomes of JEC21α and XL280α were kept for the following bioinformatics analysis. The DEGs were assessed using DEseq2 v1.16.1 of the R package and defined based on the fold change criterion ($\log_2$|fold-change| > 1.0, q value < 0.01). The gene ontologies and *P*-values of the GO terms were calculated by BiNGO v3.0.3 using a hypergeometric distribution with Benjamini-Hochberg false discovery rate (FDR) correction. In all RNA-seq assays performed in this study, two biological replicates were included.

## Statistical analysis

Statistical analyses were performed using R, version 3.4.2, and the statistical tests are indicated in the corresponding figure legends or Results section. We used two-tailed Student's *t*-test to compare the mean florescence intensity or transcript levels between two groups. Fisher's exact test was utilized to evaluate the significance of the overlap between two sets of genes. A two-sided Kolmogorov-Smirnov test was used to verify the normality of the distribution of the continuous variables. *p-values<0.05 were considered significant, and ***p-values<0.001 were considered very significant. In all figures, the error bars represented the mean ±standard deviation (SD) from at least three independent experiments.

## Acknowledgements

We thank Prof. Xiaorong Lin for critical reading and Mr. Xin Zhao, Mr. Weixin Ke and Ms. Huimin Liu for their technical assistance. This work was financially supported by the National Science and Technology Major Project (2018ZX10101004), the Key Research Program of the Chinese Academy of Sciences (QYZDB-SSW-SSMC040), the National Natural Science Foundation of China (Grants 31770163, 31622004, 31570138, 31501008 and 31501009).

## Additional information

### Funding

| Funder | Grant reference number | Author |
|---|---|---|
| Ministry of Science and Technology of the People's Republic of China | 2018ZX10101004 | Linqi Wang |
| National Natural Science Foundation of China | 31622004, 31570138, 31770163 | Linqi Wang |
| Chinese Academy of Sciences Key Project | QYZDB-SSW-SSMC040 | Linqi Wang |
| National Natural Science Foundation of China | 31501008 | Guang-Jun He |

| National Natural Science Foundation of China | 31501009 | Xiuyun Tian |
|---|---|---|

The funders had roles in study design, data collection and interpretation, or the decision to submit the work for publication.

## Author contributions

Linxia Liu, Conceptualization, Formal analysis, Validation, Investigation, Visualization, Writing—original draft; Guang-Jun He, Yingying Chen, Formal analysis, Validation, Investigation, Visualization; Lei Chen, Conceptualization, Data curation, Formal analysis, Validation, Investigation, Visualization, Methodology; Jiao Zheng, Lan Shen, Xiuyun Tian, Formal analysis, Investigation; Erwei Li, Resources, Validation, Writing—review and editing; Ence Yang, Software, Formal analysis, Visualization, Methodology; Guojian Liao, Resources, Project administration, Writing—review and editing; Linqi Wang, Conceptualization, Resources, Data curation, Formal analysis, Supervision, Funding acquisition, Validation, Visualization, Methodology, Writing—original draft, Project administration, Writing—review and editing

## Author ORCIDs

Linqi Wang (iD) http://orcid.org/0000-0002-5243-341X

## Decision letter and Author response

Decision letter https://doi.org/10.7554/eLife.38683.039
Author response https://doi.org/10.7554/eLife.38683.040

# Additional files

## Supplementary files

- Supplementary file 1. Expression profiles of various developmental stages during unisexual reproduction.

DOI: https://doi.org/10.7554/eLife.38683.032

- Supplementary file 2. Expression profiles of $PUM1^{OE}$ during unisexual reproduction.

DOI: https://doi.org/10.7554/eLife.38683.033

- Transparent reporting form

DOI: https://doi.org/10.7554/eLife.38683.034

## Data availability

The GEO accession number for the RNA-seq data reported in this study is GSE111975.

The following dataset was generated:

| Author(s) | Year | Dataset title | Dataset URL | Database and Identifier |
|---|---|---|---|---|
| Liu L, He G, Chen L, Zheng J, Chen Y, Shen L, Tian X, Li E, Yang E, Liao G, Wang L | 2018 | Genetic basis for coordination of meiosis and sexual structure maturation in Cryptococcus neoformans | https://www.ncbi.nlm.nih.gov/geo/query/acc.cgi?acc=GSE111975 | NCBI Gene Expression Omnibus, GSE111975 |

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
