## [Decision Letter]

Thank you for submitting your article "Genetic basis of meiosis-coupled developmental continuity in *Cryptococcus neoformans*" for consideration by *eLife*. Your article has been reviewed by three peer reviewers, and the evaluation has been overseen by a Reviewing Editor and Anna Akhmanova as the Senior Editor. The following individual involved in review of your submission has agreed to reveal his identity: Timothy Y. James (Reviewer #1).

The reviewers have discussed the reviews with one another and the Reviewing Editor has drafted this decision to help you prepare a revised submission.

Summary:

The manuscript by Liu et al. addresses the regulation of meiosis and sexual sporulation in *C. neoformans*. The authors discuss the "coupling" of these processes and identify key regulators, Csa1 and Csa2, that act both in the nuclear process and in the cell differentiation process. The paper and experiments are well outlined for the most part, and it is clear that the new regulators are central to the developmental program of gametogenesis in this important fungal pathogen.

Liu et al. provide an interesting perspective on the coordination of meiosis and basidium formation in the fungal pathogen *Cryptococcus neoformans*. This process is particularly important as basidiospores are believed to be the major route of infection. Previously, it was known that deletion of genes impairing meiosis does not completely eliminate basidiospore formation, and vice versa. The major contribution of this paper is defining a coupling of the two regulatory networks through two proteins Csa1 and Csa2. These proteins were found through a screen for genes upregulated in the Pum1 regulon previously implicated in hyphal growth and sexual reproduction in *C. neoformans*. The important finding is that Csa1 and Csa2 are more specific and essential for coupling meiosis and basidiospore production. Overall these results add to our understanding of how these two essential developmental processes are coordinated in *C. neoformans*. The paper is well written and the methods and analyses appear appropriate.

On a broad level, these results contribute to our understanding of developmental biology and gene regulatory networks using a simple model system of growing importance. From the detailed perspective, I found it interesting that there was no difference between unisexual and bisexual reproduction in these processes, which sheds light into the role of out-crossing in sexual development of *C. neoformans*.

The manuscript will benefit from revision to address the points in the section below. Moreover, providing a framework for studies on meiosis and sporulation in other fungi will place this in a broader context. Finally, addressing the possible functions of the two key genes and gene products identified, Csa1 and Csa2, and providing parallels to what is known about related proteins in other fungi will help broaden the scope and relevant readership for this contribution.

Essential revisions:

A major concern was not with the experiments per se but with labeling of the process as a "coupling of meiosis and basidial development". In most (if not all?) fungal species gametogenesis is the process of both (1) specialized nuclear divisions, and (2) sexual spore formation (e.g., see review by van Werven and Amon, 2011). Furthermore, in species where this has been explored these processes are co-regulated by transcriptional cascades that drive both of these processes. Therefore, highlighting a "coupling" between meiosis in the nucleus and meiospore formation or a "fused" regulation seems misleading as these events are co-regulated in fungal species. The manuscript should be revised to address this concern.

A second major concern was that the broader context for the discoveries was not made clear for researchers outside of *C. neoformans*. There was little if any discussion of meiosis in other species, or how the discoveries in *C. neoformans* would compare/contrast to the literature. This was particularly striking when it came to Csa1/Csa2 as these contain RNA recognition motifs (RRM), and studies in *S. pombe* have shown that RRM-containing proteins are key regulators of meiosis. For example, Mei2 is an RRM-containing protein and is a master regulator of *S. pombe* meiosis, and the ability to bind to RNA is crucial for its meiosis-inducing activity. Other *S. pombe* meiosis proteins also contain RRM domains such as Spo5/Mug12, which appears to be critical for progression through meiosis (Kasama et al., 2006). A discussion of points such as these seems critical to set the current findings in a much broader context, and to explain why researchers outside of those studying *C. neoformans* would be interested in the work. As such, it becomes a readership question for the current manuscript.

A minor criticism that should be addressed is that the authors went from a set of 840 Group II genes unregulated in sexual development at 24 hrs post induction to a set of 8 genes. They also had a set of related genes from Pum1 over expression that were discussed and overlap mentioned, but more detail would be useful if that is how they got to the set of 8, and the overlap of the two data sets (Pum1 over expression and time series). I get that these genes are likely transcription factors and that you can't knock out 840 genes, but where do these other co-expressed genes fit into the big picture of sexual development? I was confused about how to interpret Figure 4A and whether this was from Pum1 over expression and if so, why the Pum1 regulon was only (modestly) up-regulated. From the end of the subsection “Monitoring gene induction timing through unisexual development reveals the gene 206 network that specifically orchestrates meiosis and fruiting body development”, it seems like the data are from over-expression, but probably this isn't right.

Another question raised is that Csa1 and Csa2 are considered specific, certainly more than Pum1. Pum1 is considered pleiotropic. I didn't notice evidence that Csa1 and Csa2 were explored for other pleiotropic effects. Clearly they are able to mate efficiently.

There were a mix of serotype D and serotype A strains used for mCherry expression, development, mating and gene expression. It would be helpful to clarify:

a) Were there meaningful differences between serotypes? It was noted in Csa1/Csa2 experiments that both these genes are essential in the serotype backgrounds tested.

b) Were all RNASeq experiments performed with serotype D strains? Otherwise the alignment only to JEC21 assembly and annotation seem limiting without showing that gene expression values aren't in error due to mismatches due to the cross-species alignments?

The demonstration of a decoupling between the developmental, maturation of spores and meiotic processes is quite interesting and an important finding. However, the reader is left at the conclusion of the manuscript missing the full implications of the findings or how these conclusions are based on the observed data. This may be a reviewer and editor limitation in thinking about this system, but there are a lot of experiments and gene groupings used to infer common function or regulation but the logic throughout is not always crystal clear. Please revise to address.

There are massive data sets collected here regarding gene expression during these stages, the high level grouping of gene sets by temporal expression pattern, but how do these all connect. Figure 5B shows expression of WT Csa1 and Csa2 are only different at 24 hrs – but how does this demonstrate "This finding is suggestive of co-regulation of these two genes". Network correlation analysis is one way to examine if co-regulation is occurring and provide statistical support for such a statement.

Overall there are impressive data and reasonable interpretation but the presentation and the summary of the findings can be improved and re-framed to better support the conclusions.

---

## [Author Response]

Essential revisions:A major concern was not with the experiments per se but with labeling of the process as a "coupling of meiosis and basidial development". In most (if not all?) fungal species gametogenesis is the process of both (1) specialized nuclear divisions, and (2) sexual spore formation (e.g., see review by van Werven and Amon, 2011). Furthermore, in species where this has been explored these processes are co-regulated by transcriptional cascades that drive both of these processes. Therefore, highlighting a "coupling" between meiosis in the nucleus and meiospore formation or a "fused" regulation seems misleading as these events are co-regulated in fungal species. The manuscript should be revised to address this concern.

We modified our statements in this submission according to the suggestion of the reviewer. For instance, we have changed the title to “…Genetic basis for coordination of meiosis and sexual structure maturation in *Cryptococcus neoformans*
…” to more accurately reflect our findings. In addition, the descriptions such as “…meiosis and its fused differentiation event…” and “…coupling of meiosis and basidial differentiation…” have been replaced with the ones related to the concepts of “…meiosis and basidial differentiation…” and “…coordination of meiosis and basidial differentiation…” throughout the manuscript to avoid potential confusion.

A second major concern was that the broader context for the discoveries was not made clear for researchers outside of C. neoformans. There was little if any discussion of meiosis in other species, or how the discoveries in C. neoformans would compare/contrast to the literature. This was particularly striking when it came to Csa1/Csa2 as these contain RNA recognition motifs (RRM), and studies in S. pombe have shown that RRM-containing proteins are key regulators of meiosis. For example, Mei2 is an RRM-containing protein and is a master regulator of S. pombe meiosis, and the ability to bind to RNA is crucial for its meiosis-inducing activity. Other S. pombe meiosis proteins also contain RRM domains such as Spo5/Mug12, which appears to be critical for progression through meiosis (Kasama et al., 2006). A discussion of points such as these seems critical to set the current findings in a much broader context, and to explain why researchers outside of those studying C. neoformans would be interested in the work. As such, it becomes a readership question for the current manuscript.

We thank the reviewer for the suggestion. In the updated manuscript, we have included a new statement in the Discussion section to emphasize the conserved meiosis-inducing activity of Mei2 homologs in divergent eukaryotes. Besides, we have added new data based on a phylogenetic assay to propose the potential conserved engagement of Csa1-like and Csa2-like proteins in coordinating basidial maturation and meiotic progression in the species belonging to the Tremellales clade (new Figure 7—figure supplement 1). The corresponding description reads as follows: “In addition to Csa1 and Csa2, many RRM family members in different eukaryotic kingdoms have been reported to control sexual development or meiosis, but most of them are not similar in their protein sequences, except for Mei2 and its homologs (Jeffares et al., 2004). […] This may suggest the conserved and concerted function of these homologs in coordinating basidial maturation and meiotic progression in Tremellales”.

A minor criticism that should be addressed is that the authors went from a set of 840 Group II genes unregulated in sexual development at 24 hrs post induction to a set of 8 genes. They also had a set of related genes from Pum1 over expression that were discussed and overlap mentioned, but more detail would be useful if that is how they got to the set of 8, and the overlap of the two data sets (Pum1 over expression and time series).

We agree with the reviewers that more detail regarding the transcriptome-guided identification of the set of candidate regulator genes would be ideal. We have revised the corresponding statements to address this point: “Considering the temporal overlap of meiosis and basidial differentiation, we hypothesized that the genes dedicated to the coordination of these two events are likely included in meiosis gene-enriched group II. […] RNA-seq analysis identified eight group II regulatory genes (Figure 4C) that displayed remarkably induced expression in response to Pum1 overexpression”.

I get that these genes are likely transcription factors and that you can't knock out 840 genes, but where do these other co-expressed genes fit into the big picture of sexual development?

To address this concern, we have added two new descriptions into the Results and Discussion sections, respectively, to extensively discuss the functional involvement of the various biological processes mediated by group II genes during fruiting body development in *C. neoformans* and other basidiomycetes:

Results section:

“The GO analysis identified various GO terms in these sub-groups and, as expected, revealed enrichment of meiosis genes. […] These processes mainly include rRNA/protein metabolic processes, which may reflect cellular protein turnover in response to stress after extended inoculation on V8 juice agar, a relatively nutrition-restrictive medium.”

Discussion section:

“Profiling gene induction during mating differentiation further unveiled a special gene group (group II) potentially responsible for the coordination of these two processes (Figure 3B). […] This may be indicative of the conserved importance of fatty acid metabolism during fruiting body development in divergent fungi.”

I was confused about how to interpret Figure 4A and whether this was from Pum1 over expression and if so, why the Pum1 regulon was only (modestly) up-regulated.

We apologize that our description in the original manuscript may have been confusing and may have been misleading to the reviewer. Figure 4A illustrates the enrichment of the genes belonging to different signaling cascades in the four gene groups (group I, group II, group III and group IV), which were defined based on RNA-seq-guided analysis of gene induction timing throughout unisexual development (Figure 3B). The signaling genes used in this enrichment evaluation have been integrated into Figure 4—source data 2 for clarification. These genes include those encoding experimentally verified elements (Mating MAPK pathway) or the genes activated by the activators dominating different sexual stages (Mat2, Znf2 or Pum1). The genes activated by Mat2 and Znf2 (“Mat2-activated genes” and “Znf2-activated genes”) are derived from the previous transcriptome data (Lin et al., 2010), and the gene set “Pum1-activated genes” is generated based on the RNA-seq analysis of the *PUM1* overexpression strain in this study (Supplementary file 2). This information has been described in the caption of Figure 4A in this version of the manuscript.

Gene enrichment assessment indicated that only the set of genes activated by Pum1 was significantly over-represented in group II but not in the other gene groups (group I, *P* = 0.37; group II, *P* = 3.95×10^-7^; group III, *P* = 0.14; group IV, *P* = 1.0; Fisher test) (Figure 4A). Conversely, the genes belonging to other signaling cascades display either the highest enrichment in group I (mating MAPK pathway genes and Mat2-activated genes), which mainly includes the genes responsible for early and middle mating events, or a similar level of enrichment between group I and group II (Znf2-activated genes). The specific enrichment of the Pum1-activated regulon in group II supports its key role in late sexual stages, including meiosis and basidial formation. Gene Ontology analysis further confirmed that many Pum1-activated group II genes are involved in meiosis (Figure 4B). In the revised manuscript, we have now redrawn Figure 4A and revised the corresponding description in its caption to highlight the specific enrichment of Pum1-activated genes in gene group II.

From the end of the subsection “Monitoring gene induction timing through unisexual development reveals the gene 206 network that specifically orchestrates meiosis and fruiting body development”, it seems like the data are from over-expression, but probably this isn't right.

The reviewer is correct that the description in the original version of the manuscript is related to the targets activated by Pum1, which were revealed via RNA-Seq profiling targeting the *PUM1* overexpression strain. To clarify this point, we have added more information regarding how these data were generated to the legends of Figures 4A and 4B, which show the results mentioned in original version of the manuscript.

Another question raised is that Csa1 and Csa2 are considered specific, certainly more than Pum1. Pum1 is considered pleiotropic. I didn't notice evidence that Csa1 and Csa2 were explored for other pleiotropic effects. Clearly they are able to mate efficiently.

Our previous study has indicated that in addition to meiosis and basidial maturation, Pum1 also plays an important role in other mating processes. For instance, Pum1 inhibits the expression of early mating genes, such as *MFα* (α pheromone synthesis gene), while promoting the middle mating event filamentation (Wang et al., 2014). By comparison, Pum1 targets Csa1 and Csa2 are dispensable for self-filamentation (Figure 4—figure supplement 1) and appear to control meiotic progression and fruiting body differentiation in a more specific manner. To further corroborate this idea, we have performed new phenotypic assays to quantitatively assess the influence of Csa1 and Csa2 on filamentous initiation during unisexual development (new Figure 4—figure supplement 2). As expected, no evident defect during this stage was detected in the absence of either Csa1 or Csa2. In contrast, deletion of *PUM1* adversely affected the initiation of self-filamentation (new Figure 4—figure supplement 2). Likewise, disruption of *PUM1* but not *CSA1* and *CSA2* severely attenuated bisexual filamentation (new Figure 5—figure supplement 1). The RT-PCR analysis further demonstrated that deletion of *CSA1* or *CSA2* cannot significantly change the expression of the genes involved in early (*MFα* and *MAT2*) or middle mating stages (*ZNF2*), respectively, and their upstream regulatory gene *PUM1* (please see Author response image 1). These new results strongly support that *CSA1* and *CSA2* play a more specific role than Pum1 in orchestrating meiosis and basidial maturation in *C. neoformans*.

**Author response image 1. respfig1:** RT-PCR analysis showed that deleting either or both of *CSA1* and *CSA2* cannot significantly change the mRNA levels of *MFα, MAT2, ZNF2* and *PUM1* at 24 hrs post unisexual mating stimulation. Bars show the mean ± SD of three replicates.

There were a mix of serotype D and serotype A strains used for mCherry expression, development, mating and gene expression. It would be helpful to clarify:a) Were there meaningful differences between serotypes? It was noted in Csa1/Csa2 experiments that both these genes are essential in the serotype backgrounds tested.

The serotype D strain XL280 was chosen for most experiments performed in this study because of its well-described robust ability to undergo sexual development (Lin et al., 2006, Zhai et al., 2013). This aspect enabled us to sensitively assess the various phases during the sexual cycle in certain mutants. To test whether Csa1 or Csa2-activated sporulation is unique to the XL280α background, their coding genes were mutated in JEC21 (serotype D) and H99 (serotype A), respectively. We showed that *CSA1* and *CSA2* are essential for bisexual sporulation in both serotype A and serotype D strains, which belong to the *Cryptococcus neoformans* species complex and diverged from each other for at least 18.5 million years (Xu et al., 2000). These data demonstrated that the requirement of Csa1 and Csa2 for the formation of meiospores is not limited to XL280, the model isolate used for studying *Cryptococcus* sex, but is conserved among the strains in the *C. neoformans* species complex. We have modified the corresponding description to address this issue: “To test whether *CSA1*-activated or *CSA2*-activated sporulation is unique to the XL280 (serotype D) background, *CSA1* and *CSA2* were individually mutated in the JEC21 (serotype D) and H99 (serotype A) backgrounds. […] These data demonstrated that the requirement of Csa1 and Csa2 for the formation of meiospores is not limited to XL280 but is conserved among strains in the *C. neoformans* species complex…”.

b) Were all RNASeq experiments performed with serotype D strains? Otherwise the alignment only to JEC21 assembly and annotation seem limiting without showing that gene expression values aren't in error due to mismatches due to the cross-species alignments?

The reviewer is correct that all RNA-seq experiments in this study were carried out using the serotype D isolate XL280 (XL280α). RNA-seq reads were first mapped to the genome sequence of XL280 to determine the unigenes. These unigenes were further aligned against well-annotated genome sequence of another *C. neoformans* isolate JEC21α to reduce false positives due to variation in gene prediction process. The protein coding genes found in both genomes of JEC21α and XL280α were kept for the following bioinformatics analysis, as our aim is to identify the conserved gene determinants underlying the orchestration of meiosis and basidial differentiation among isolates in the *C. neoformans* species complex but not those specific to a given genetic background. In the revised manuscript, we have added two new descriptions to clarify the RNA-seq experiments performed in this study (the Results: subsection “Monitoring gene induction timing during unisexual development reveals the gene network orchestrating meiosis and basidium development”, first paragraph; the Materials and methods: subsection “RNA-seq and data analysis”).

The demonstration of a decoupling between the developmental, maturation of spores and meiotic processes is quite interesting and an important finding. However, the reader is left at the conclusion of the manuscript missing the full implications of the findings or how these conclusions are based on the observed data. This may be a reviewer and editor limitation in thinking about this system, but there are a lot of experiments and gene groupings used to infer common function or regulation but the logic throughout is not always crystal clear. Please revise to address.There are massive data sets collected here regarding gene expression during these stages, the high level grouping of gene sets by temporal expression pattern, but how do these all connect.

We thank the reviewers/editor for these helpful suggestions. To clarify the logical connection between the observed data and the conclusions drawn in the manuscript, we have revised the corresponding descriptions, particularly those related to how the key regulatory determinants connecting meiosis and basidial differentiation were identified through a combination of transcriptomic approaches and quantitative phenotypic assays (the Results: subsection “Monitoring gene induction timing during unisexual development reveals the gene network orchestrating meiosis and basidium development”). We have also redrawn Figure 4A and added more detail to its caption to provide a better understanding of the finding that the Pum1-actived regulon is specifically enriched in gene group II. Besides, we have performed new transcriptional experiments along with the appropriate statistical assessment (new Figures 5B and 5C) to corroborate the conclusion of co-upregulation of *CSA1* and *CSA2* by Pum1 (please see our detailed response below). To improve the implications of the findings, new statements have been included to propose the potential metabolic or molecular involvement in supporting fruiting body differentiation in *C. neoformans* and other basidiomycetes, based on the results revealed by profiling gene induction during mating differentiation (the Results: subsection “Monitoring gene induction timing during unisexual development reveals the gene 209 network orchestrating meiosis and basidium development”; the Discussion: second paragraph). Moreover, we have integrated the phylogenetic data into new Figure 7—figure supplement 1 to support that *CSA1-like* and *CSA2-like* genes co-exist in the genomes of species belonging to Tremellales, in which they may play a conserved role in the coordination of meiosis and basidial maturation.

Figure 5B shows expression of WT Csa1 and Csa2 are only different at 24 hrs – but how does this demonstrate "This finding is suggestive of co-regulation of these two genes". Network correlation analysis is one way to examine if co-regulation is occurring and provide statistical support for such a statement.

We agree with the reviewer that the conclusion regarding the co-regulation of *CSA1* and *CSA2* needs be experimentally and statistically confirmed. To address the concern, we have performed new RT-PCR experiments at seven time points after mating induction (6hrs, 12hrs, 15hrs, 18hrs, 24hrs, 48hrs and 72hrs) to generate the detailed expression dynamics of *CSA1* and *CSA2* during unisexual development (new Figure 5B). The results of the Pearson’s test indicated that *CSA1* and *CSA2* shared a very similar temporal expression pattern (new Figure 5B, *r* = 0.96, *P* = 6.4 × 10^-5^, Pearson’s test). These data support that Csa1 and Csa2 may be co-regulated during unisexual reproduction. Confirming this hypothesis, our transcriptional evidence based on RT-PCR evaluation demonstrated that these genes are co-upregulated by Pum1 (new Figure 5C).

Additional references:

Kasama T, Shigehisa A, Hirata A, Saito TT, Tougan T, Okuzaki D, Nojima H. 2006. Spo5/Mug12, a putative meiosis-specific RNA-binding protein, is essential for meiotic progression and forms Mei2 dot-like nuclear foci. Eukaryot Cell. 2006 Aug;5(8):1301-13.